# Robust Silent Localization of Underwater Acoustic Sensor Network Using Mobile Anchor(s)

**DOI:** 10.3390/s21030727

**Published:** 2021-01-21

**Authors:** Rahul Mourya, Mauro Dragone, Yvan Petillot

**Affiliations:** Institute of Sensors, Signals, and Systems, Heirot-Watt University, Edinburgh EH144AS, UK; M.Dragone@hw.ac.uk (M.D.); Y.R.Petillot@hw.ac.uk (Y.P.)

**Keywords:** acoustic sensor network, TDoA, network localization, robust estimation, mobile anchor

## Abstract

Underwater acoustic sensor networks (UWASNs) can revolutionize the subsea domain by enabling low-cost monitoring of subsea assets and the marine environment. Accurate localization of the UWASNs is essential for these applications. In general, range-based localization techniques are preferred for their high accuracy in estimated locations. However, they can be severely affected by variable sound speed, multipath spreading, and other effects of the acoustic channel. In addition, an inefficient localization scheme can consume a significant amount of energy, reducing the effective life of the battery-powered sensor nodes. In this paper, we propose robust, efficient, and practically implementable localization schemes for static UWASNs. The proposed schemes are based on the Time-Difference-of-Arrival (TDoA) measurements and the nodes are localized passively, i.e., by just listening to beacon signals from multiple anchors, thus saving both the channel bandwidth and energy. The robustness in location estimates is achieved by considering an appropriate statistical noise model based on a plausible acoustic channel model and certain practical assumptions. To overcome the practical challenges of deploying and maintaining multiple permanent anchors for TDoA measurements, we propose practical schemes of using a single or multiple surface vehicles as virtual anchors. The robustness of localization is evaluated by simulations under realistic settings. By combining a mobile anchor(s) scheme with a robust estimator, this paper presents a complete package of efficient, robust, and practically usable localization schemes for low-cost UWASNs.

## 1. Introduction

With advances in wireless communication and networking technologies, power-efficient edge-computing devices, miniaturization of sensor payloads, and implementations of advanced signal processing algorithms and machine learning inferences on edge-computing devices, the Internet of Things (IoT) has taken the world by storm: from home automation to industrial automation to environmental monitoring. One of the significant achievements of such sensor networks is remote monitoring and controlling of processes in hazardous environments without human interventions for long-term. In the recent decade, similar initiatives have raised strong interest in the subsea domain, which covers two-thirds of our planet and is one of the main sources of resources. Underwater sensor networks have the potential to revolutionize the monitoring of subsea assets and marine environments, and enhance the prevention of disasters, security, and navigation. However, compared to terrestrial IoT, underwater wireless sensor networks are lagging due to the challenging nature of the subsea environment. The two key challenges are the limited underwater communication channels and power supply to the sensor nodes for long-term deployment. The widely used electromagnetic and optical channels in the terrestrial environment are restricted to very short distances in an underwater environment [1,2]. Therefore, acoustics are predominantly used, which can travel up to a few kilometers without significant power loss [3]. However, the acoustic propagation in the ocean suffers from attenuation that increases with signal frequency, time-varying multipath propagation, and low sound speed (typically 1500 m/s), which effectively results in narrow communication bandwidth up to only a few tens of KHz [1,4]. Currently, the sensor nodes are battery operated and have a limited power budget. Sending data across the network is the major contributor to energy consumption.

Nevertheless, recent developments in acoustics modeling, modem technology, and the adoption of modern digital modulation detection techniques have enabled reliable underwater communication over a long-range at a usable data rate. Today, many commercially available acoustic modems costing from a few hundred to a few thousands of dollars can communicate from a few meters up to ten kilometers with a data rate from 1.5 to 30 kbps (the higher rate at a shorter range, and vice versa) (https://evologics.de/acoustic-modems; https://www.sonardyne.com/product/underwater-acoustic-modems). Their power consumption ranges from 0.5 watts (in receive mode) up to 60 watts (in transmit mode). Recently, as part of the EPSRC UK funded USMART project, Newcastle University, UK, has developed a low-cost low-power acoustic modem, called the Nanomodem (https://research.ncl.ac.uk/usmart), which is small in size (42 mm diameter × 60 mm height), has a weight less than 200 g, and costs under 60 dollars. It can communicate reliably over 3 km in a moderately noisy channel at a data rate of 640 bps while consuming only 12 mW in listening, 25 mW in receiving, and 1.5 W in transmitting modes, respectively. Similar advances in sensor technology have produced low-cost miniature size sensor payloads. Advances in system-on-chip technology have produced ultra-low powered edge computing devices with huge computational power. Similar significant progress has been made in network protocols including several acoustically aware medium-access-control (MAC) protocols [5,6,7]. Together, these advances have made the prospect of affordable large-scale UWASNs for many practical applications.

In order to establish sustainable UWASNs for long-term applications, the major tasks are deployment of the sensor nodes in the region-of-interest (ROI), localization of the nodes, and setting up network protocols for reliable communication across the network. More importantly, to be able to use such networks extensively in different applications, the initial investment and the maintenance cost should be low. The challenging environment of the subsea domain, the highly variable nature of the acoustic channels, the narrow bandwidth, and the limited power supply pose extra difficulties in establishing UWASNs when compared to terrestrial IoT. In this paper, we focus on the localization task of static (slowly drifting) UWASNs. We will discuss in detail the difficulties associated with it, and then present robust localization schemes that are theoretically sound and practically implementable on low-cost and low-power UWASNs.

### 1.1. Network Localization: Challenges and Related Works

Deployment and accurate localization of the UWASNs are critical for the sustainable operation of the network for many applications, e.g., spatio-temporal monitoring, target detection, and network management [8], and it can be increasingly challenging for large-scale network. For our purpose, we consider three different scales of UWASNs: (i) a few tens (10–50) of nodes spread over an area up to 9–12 km${}^{2}$ as a small-scale network, (ii) a few tens to hundreds (50–200) nodes spread over an area up to 12–36 km${}^{2}$ as a mid-scale network, and (iii) a few hundred to thousands (200–1000) nodes spread over an area up to 36–100 km${}^{2}$ as a large-scale network.

Deploying and maintaining sensor nodes precisely at predefined locations may not be a viable option except in the case of a small-scale network. A practical approach, often used, is to drop them from the surface and they settled down approximately at the planned locations. Thus, their exact locations are not generally known, especially in the deepsea scenario. Moreover, they do drift slowly over time after deployment. Thus, it becomes essential to employ a reliable and efficient localization scheme to localize the nodes periodically, e.g., every day or every week, depending upon the turbulence in the ROI.

Several localization schemes are proposed in the literature depending on techniques to measure physical parameters (e.g., range and angle) between the nodes, topologies of the network, and localization accuracy requirements. Localization schemes can be classified broadly into two categories: anchor-based and anchor-free localization, see [9,10,11] for a broad survey on localization schemes. Anchor-based localization schemes are used when the absolute locations of the sensor nodes are required, whereas anchor-free localization schemes are preferred when only relative locations are sufficient. Anchor-based localization requires ranges, or range-difference, and/or the angles between the unknown node and the anchors. These quantities are measured by one of the following techniques: (i) Received Signal Strength Indicator (RSSI), (ii) Angle-of-Arrival (AoA), (iii) Time-of-Arrival (ToA), or (iv) Time-Difference-of-Arrival (TDoA), see [12] for details on each of them. AoA requires extra hardware to measure the direction of the acoustic signal, whereas the other two techniques need only an acoustic modem with a precise clock. Thus, range-based techniques are commonly preferred in a low-cost system and they require a minimum number of anchors: in *n*-dimensional space it requires at least n+1 anchors. Given the locations of the anchors and the ranges (or range-differences), the unknown location of a node is estimated by a technique called Trilateration (or Multilateration), which, in its simplest form, finds common intersections of circles (or hyperbolas) [12]. ToA and TDoA-based techniques generally provide higher accuracy for range and range-difference estimation, respectively. RSSI can be used to complement ToA and TDoA-based techniques. ToA between two nodes can be measured by one-way pinging when the clocks of the two nodes are precisely synchronized and two-way pinging when synchronization is not possible or inaccurate. The latter technique is usually preferred because clock synchronization between two low-cost nodes is practically difficult as it may require expensive hardware as well as time and energy-consuming synchronization protocols. Similarly, TDoA can be performed with or without clock synchronization among the nodes, but for the same reasons, TDoA without clock synchronization is preferable. An efficient scheme for measuring TDoAs, which does not require any clock synchronization and two-way pinging among the anchors and the sensors, was proposed in [13] for terrestrial wireless sensor networks. Later, it was adapted for the underwater domain in [14] and was referred to as Silent Underwater Positioning Scheme (UPS). UPS saves both the communication bandwidth and the energy of the sensors and scales with the size of the network; the network size is only limited by the availability of anchors and their acoustic ranges. In this paper, we consider the UPS for TDoA measurements for the proposed robust localization schemes.

In general, locations estimated by multilateration suffer from errors in range-difference measurements due to the following reasons: (i) delay in the receiver system, (ii) variable sound speed, and (iii) time-varying multipath propagation of acoustics in oceans [14]. The receiver system delay is the duration between the instant the signal hits the receiver and the signal reception is registered by it. This time delay is determined by the receiver electronics, and it can be constant or approximated as white Gaussian noise. Sound speed variation is a consequence of spatial variability of salinity, temperature, and pressure [15]. Due to variable sound speed, an acoustic signal may arrive at the receiver earlier or later than the calculated time assuming constant sound speed, e.g., 1500 m/s, in seawater. It can introduce uncertainties in the range-difference measurements, which are not easy to model statistically.

When a source emits an acoustic wave, also seen as a beam of rays, each ray follows a slightly different path and a receiver placed at a certain distance observes multiple signal arrivals with different strengths, phases, and delays. This phenomenon is referred to as multipath propagation and is governed by two effects: sound reflection at the surface, bottom, and any objects, and refraction due to variable sound speed [1,3]. Figure 1 depicts multipath propagation in both the shallow and the deepsea scenarios and variation of sound speed with depth. An adverse effect of multipath is the uncertainty in the range-difference measurements due to the problem of detecting the first direct path among multiple closely packed indirect paths arriving at the receiver. Another consequence is intersymbol interference at the receiver, which leads to high error rates in symbol detection [16,17]. Modeling the uncertainty in the measured range-difference due to variable sound speed and the multipath propagation is a difficult problem and so far no statistical model is proposed to characterize it accurately.

Several efforts have been made to model channel properties under different environmental conditions, which can be categorized into two classes: deterministic and statistical models. Deterministic models use physics to characterize the acoustic rays within certain boundary conditions, variable sound speed, and other parameters, whereas the statistical models focus on estimating overall probability distributions of the channels under different conditions. The Bellhop ray tracing model developed in [18,19] is a commonly used deterministic model for studying acoustic propagation in a fixed environment. It has been extended in [20] to handle limited variations in the environment. The wavefront model [21] is another deterministic approach that provides an approximation to the ray theory to model the effect of the curvature of the surface waves, amplitude, and arrival time fluctuations. See the work in [22] for comparison between different deterministic models. Several stochastic channel fading models are proposed based on the experimental acoustic data collected at different geographical locations and seasons. Some researchers found their observations fitting to Rayleigh fading [23,24,25], while some found it to be Rician fading [26,27,28], and some found it to be K-distribution [29,30]. This variation is due to the highly dynamic nature of the ocean, which includes both small- and large-scale phenomena [31].

The statistical models mentioned above are not directly useful in deducing the statistical model of the errors in range/range-difference measurements due to multipath propagation. In this regard, the modified ultra-wideband Saleh–Valenzuela (UWB S-V) model [32,33] is useful. In simple words, this model states multipath contributions generated by a signal arrives as clusters of multiple rays following a Poisson process with a certain arrival rate and within each cluster, the rays follow an another Poisson process with a different arrival rate. With certain adaptions, as discussed in Section 4, we believe that the modified UWB S-V model is fairly suitable for describing the multipath acoustic channel. Based on the model, errors in ToA measurements can be modeled by an Exponential distribution and, thus, the errors in TDoA measurements by a Laplacian distribution. With other certain assumptions, as discussed in Section 5, we deduce that the errors in range-difference measurements can be appropriately modeled by an independent and identically distributed (i.i.d.) Laplacian distribution, which has possibilities of extreme values, i.e., outliers in the measurements. The presence of outliers in the measurements has a significant adverse effect on the estimated locations. Thus, robust estimators, which can identify and avoid outliers during the estimation process, are essential for the localization of UWASNs.

The literature on robust localization for UWASNs is preliminary and limited in numbers compared to the terrestrial wireless networks owing to the challenging environment of the oceans and the UWASNs are relatively a new research field. In [14], the authors consider a crude way of dealing with the problem by averaging the TDoA measurements performed over several cycles. Many others [34,35,36] are motivated by approaches based on multipath propagation of terrestrial radio signals and propose computationally expensive algorithms to distinguish between line-of-sight and non-line-of-sight rays due to only obstacles. In [37], a robust joint localization and synchronization in UWASNs is presented based on two-way pinging between anchors and the sensors; thus, it is suitable only for small-scale networks and is power-consuming. To the best of our knowledge, there is no research paper on robust localization based on an appropriate statistical model of errors due to variable sound speed and multipath propagation and proposes computationally efficient robust estimators.

### 1.2. Objectives and Contributions

Localizing large-scale UWASNs is a challenging task for several reasons as pointed out in the previous sections. Range-difference measurements are highly susceptible to errors, which can severely deteriorate the estimated locations if not handled properly. Thus, the first aim of this paper is to propose robust location estimators that are computationally efficient on low-cost low-power static (slowly drifting) sensor nodes. Robust estimators require multiple anchors for accurate and reliable location estimates. However, deploying and maintaining multiple permanent anchors in an ocean environment is often costly and impractical for several reasons. Thus, the second aim is to propose practical and scalable schemes for localizing a mid-scale to large-scale UWASNs using single or multiple autonomous surface vehicles acting as multiple anchors.

In this paper, we present following several incremental and novel contributions toward the stated objectives, which cover both the theoretical and the practical aspects:In general, range-based localization is a non-convex optimization problem, i.e., not easy to find global solution, as discussed in Section 2. We present guidelines to deploy the anchors in certain feasible geometrical configurations that guarantees to find the correct solution using simple (computationally less expensive) solvers. Furthermore, we present two simple solvers and compare their performances to suggest when to use one over others.To devise a robust location estimator, an appropriate model of the measurement errors is essential. We advocate that the modified UWB S-V model with certain adaptations is a fairly suitable statistical model for characterizing the acoustic multipath propagation. Using the modified UWB S-V model with certain practical assumptions, we propose that the errors in range-difference measurements can be appropriately modeled by an i.i.d. Laplacian distribution, which has possibilities of outliers in the measurements.To mitigate the errors, we present three robust location estimators: Least-Absolute-Deviations (LAD), Least-Median-Squares (LMedS), and RANdom Sample Consensus (RANSAC), which appropriately use the two simple solvers to keep computational expenses low. We perform rigorous simulations under relevant noise settings to compare the performance of the three robust estimators. Based on the simulation results, we present guidelines to select one of the robust estimators over others depending upon the noisy environment and computational budget of the sensors.To avoid deployment and maintenance of multiple permanent anchors required for robust location estimation, we propose three practical schemes to perform TDoA measurements using a combination of a static and a mobile anchor or a single mobile or multiple mobile anchors. The schemes are adaptable to the scale of the networks, resources available at hand, and required robustness in the estimated locations.Combining a mobile anchor scheme with a robust location estimator, we propose a complete package of an efficient, robust, and practically usable localization scheme for low-cost low-power UWASNs.

### 1.3. Structure of Paper

The rest of the paper is organized as follows. Section 2 presents TDoA-based localization problem from an estimation theory perspective, whereas Section 3 present in detail the UPS and numerical methods to solve the localization problem. Section 4 presents a statistical model to characterize multipath propagation in underwater acoustic channel and using it Section 5 proposes statistical model of the errors in TDoA measurements. Based on the statistical model of the errors Section 6 proposes three robust location estimators. Section 7 proposes the localization schemes using mobile anchor(s). To evaluate and compare the robust estimators, Section 8 presents rigorous simulations and results with the possible interpretations. Section 9 summarizes the whole paper and establish connections between the different sections, and then present guidelines on selecting a robust estimator depending upon the situation. Finally, Section 10 concludes the paper with possible future research directions.

### 1.4. Notations

In this paper, we denote column vectors and matrices by boldface alphabets in lowercase and uppercase, respectively, and scalar values by normal alphabets (including Greek) in lowercase. For the transpose and inverse operators acting on matrices, we use the notations ${\left(\cdot \right)}^{T}$ and ${\left(\cdot \right)}^{-1}$, respectively. We use uppercase calligraphic alphabets such as S to represents a set and |·| to represent the cardinality of a set or absolute value of a scalar depending the input argument. We denote the Manhattan norm and Euclidean norms of a vector by ${∥\cdot ∥}_{1}$ and ${∥\cdot ∥}_{2}$, respectively, while $\mathrm{diag}(a_{1},\cdots ,a_{n})$ to represent n×n diagonal matrix with $a_{i}$ be the *i*th diagonal element. We denote the expectation and variance operators acting on a random variable by E[·] and Var[·], respectively. Last, for the set of a real number, we use the notation R.

## 2. TDoA-Based Localization: Problem Formulation

Before we present the UPS scheme to measure TDoAs in the next section, first, we present the TDoA-based localization problem from the estimation theory point of view and then discuss possible approaches to solve it. Let $x\in R^{n}$, usually n=2or3, be the unknown location and $a_{i}\in R^{n},\forall i=0,\cdots ,N$ be the known locations of N+1 anchors among which the 0th index represents a reference anchor. Let $t_{i},\forall i=0,\cdots ,N$ be the pairwise ToAs between x and $a_{i}$. Then, the TDoA measured at x between the reference anchor and the remaining anchors are given by $\Delta t_{i}=t_{0}-t_{i},\forall i=1,\cdots ,N$. Assuming the sound speed be *v*, the range-differences can be written as the function of x as
$$\Delta r_{i}\left(x\right)=v\Delta t_{i}=∥x-a_{0}{∥}_{2}-{∥x-a_{i}∥}_{2},\forall i=1,\cdots ,N.$$

Let $\Delta r\in R^{N}$ be a vector formed by stacking all the range-differences $\Delta r_{i}$. The TDoA measurements are often a noisy process, thus the range-difference vector can be written as $\Delta \tilde{r}=\Delta r\left(x\right)+\varepsilon$, where ε represents an additive noise vector. In general, the randomness in $\Delta \tilde{r}$ is fully characterized by the conditional pdf of $\Delta \tilde{r}$ given x, denoted by $f(\Delta \tilde{r}|x)$. As a function of $\Delta \tilde{r}$, $f(\Delta \tilde{r}|x)$ is a measure of how likely it is to observe $\tilde{\Delta r}$, for a given value of x. Therefore, commonly known as likelihood function of $\Delta \tilde{r}$, given x. Under this interpretation, an intuitive estimate of x, from a collection of observations $\Delta \tilde{r}$, is the value that maximizes the likelihood function, which is known as the *maximum-likelihood* estimate (MLE) of x expressed as
(1)$${\hat{x}}_{MLE}=\underset{x}{arg max}f\left(\Delta \tilde{r}|x\right)$$

For the case when the range-difference measurements follow a Gaussian process, i.e., $\Delta \tilde{r}\left(x\right)$ is a Gaussian random vector with mean Δr(x) and covariance matrix $\Sigma =\mathrm{diag}(\sigma_{i}^{2},\cdots ,\sigma_{N}^{2})$, where $\sigma_{i}^{2}$ represent the variance in each measurement, then the MLE reduces to solving the following optimization problem, which is usually referred to as weighted least-square (LS) minimization problem:(2)$${\hat{x}}_{MLE}=\underset{x}{arg min}\;\sum_{i=1}^{N}\frac{{\left(\Delta \tilde{r_{i}}-(∥x-a_{0}{∥}_{2}-∥x-a_{i}{∥}_{2})\right)}^{2}}{\sigma_{i}^{2}}$$

The Gaussian distribution is not the only possibility, but the measurement errors can come from some *long-tailed* distributions process that has the possibilities of seeing some extreme values. A canonical example is the Laplacian distribution, which is characterized by two parameters: *location* and *scale* analogous to the mean and variance of the Gaussian distribution, respectively. In Section 5, we will see that the errors in range-difference measurements approximately follow an i.i.d. Laplacian distribution. Under such a noise condition, we can write the MLE of an unknown location as
(3)$${\hat{x}}_{MLE}=\underset{x}{arg min}\sum_{i=1}^{N}\;\left|\Delta {\tilde{r}}_{i}-(∥x-a_{0}{∥}_{2}-∥x-a_{i}{∥}_{2})\right|$$

The above problem (Equation 3) is commonly known by the name *least-absolute-deviations* (LAD), and its solution is robust to outliers and highly recommended over LS method when there are chances of outliers [38,39].

In general, any estimate $\hat{x}$ of x is a function of $\Delta \tilde{r}$, thus it is a random vector whose moments can be defined. The estimate $\hat{x}$ is said to be unbiased estimate of x if the condition: $E[\hat{x}-x]=0$ is satisfied. For an unbiased estimate $\hat{x}$ of x, its covariance matrix is bounded below by inverse of the Fisher information matrix, which is also commonly known Cramer–Rao Lower Bound (CRLB), and is met with equality by ${\hat{x}}_{MLE}$, when the size of the observation vector $\Delta \tilde{r}$ tend to *∞* [40].

The above problems (Equation 2) and (Equation 3) are non-convex in nature; thus, finding their global solutions is a difficult task. The simplest approach is to find the local optimum by using a gradient-based iterative method [41] starting with a good initial guess of the solution with a hope to find the global optimum. Other common approaches are based on convex-relaxation techniques, which convert the above problems into convex problems. Different convex-relaxations are suggested in the literature; some solve the problems approximately [42], and some targets to find the global optimum [43,44]. In [45,46,47], the authors consider square-range-based LAD and LS estimators, which are suboptimal in the maximum-likelihood sense, and propose Semidefinite Relaxation and Second-order cone programming relaxation, which find approximate solutions. In [48], authors consider both ToA and TDoA-based location estimation as square-range and square-range-difference LS problems, which are also suboptimal and non-convex, but propose procedures to find global solutions in many cases.

All the approaches mentioned above for solving either problem (Equation 2) or (Equation 3) or related formulations are based on computationally expensive iterative solvers. As mentioned in Section 1.2, we target to localize low-power sensor nodes. Thus, we favor computationally less demanding approaches, which either solve the suboptimal formulations or find only the local optimum of the above problems. However, we point out some practical guidelines, e.g., geometrical configuration of the anchors, which can avoid the caveats of the non-convexity and help in finding a correct solution of the problem (Equation 2) or (Equation 3) easily. In the next section, we present two such solvers, which are computationally less demanding and can produce sufficiently accurate location estimates.

## 3. Silent Underwater Positioning Scheme

Consider a sensor node deployed at an unknown location $x=\left[x,y,z\right]\in R^{3}$ in a ROI. We consider that the sensor node has a pressure gauge that provides sufficiently accurate depth measurement. This avoids deployment and maintenance of anchors deep below the sea surface, which is practically a difficult task to perform 3D-localization. Thus, *z* is a known quantity and the localization boils down to 2D-space. Let us consider N+1 anchors on the sea surface over the ROI. Without loss of the generality, we assume that all the anchors have equal and omnidirectional acoustic range. One of the anchors is chosen as a lead and others as assistants. To have maximum acoustic coverage over the ROI, we deploy the assistant anchors on a circle and the lead anchor at the center as shown in Figure 2. The locations of anchors are assumed to be known sufficiently accurately via Global Positioning System (GPS). Let $a_{i}={\left[x_{i},y_{i},z_{i}\right]}^{T}\in R^{3},\forall i=0,\cdots ,N$ represent the locations of the anchors where 0th index represents the lead anchor and remaining the assistant anchors. All the anchors and the sensor have precise clocks (no need to synchronize them). The lead anchor initiates the silent localization process, which is then followed by the assistant anchors. The whole process can be divided into two phases: (i) Range-Difference measurement phase and (ii) Multilateration phase, as described below.

### 3.1. Range-Difference Measurement Phase

The lead anchor starts the timing measurement process by broadcasting its location and delay information, collectively called a beacon signal. Let the beacon signal from the lead arrives at the sensor at $t_{0}$. The delay information from the lead anchor is zero. Now, the first assistant anchor broadcasts its beacon signal after a time delay $\delta_{1}$ measured with reference to the time at which it receives the beacon signal from the lead anchor. The beacon signal contains the tuple $\left(a_{1},\delta_{1}\right)$, and let it arrives sensor node at $t_{1}$. Similarly, the process continues until the last assistant anchor broadcasts its beacon signal. Let the delays be $\delta_{i},\forall i=1,\cdots ,N$ and the time of arrival of the beacon signals at the sensor be $t_{i},\forall i=0,1,\cdots ,N$. Now, the TDoAs at the sensor can be calculated as $\Delta t_{i}=t_{i}-t_{0},\forall i=1,\cdots ,N$. Note that all the times are measured locally so no need for clock synchronization among the anchors and the sensor. If $r_{i}={∥x-a_{i}∥}_{2},\forall i=0,\cdots ,N$ represent the range between the sensor and the anchors, and $\Delta a_{i}={∥a_{0}-a_{i}∥}_{2},\forall i=1,\cdots ,N$ represent the ranges between the lead and the assistant anchors, then the range-differences between the anchors calculated at the sensor can be written as
(4)$$\Delta r_{i}=r_{0}-r_{i}=\Delta a_{i}+v\cdot \left(\delta_{i}-\Delta t_{i}\right).$$

Note that there are three instances in this range-difference measurement process when errors due to multipath propagation and varying sound speed can get into it. These instances are when (i) the lead anchor’s beacon signal arrives at the sensor, i.e., in $t_{0}$, (ii) the lead anchor’s beacon signal arrives at the assistant anchors, i.e., in $\delta_{i}$, and (iii) the assistant anchor’s beacon signals arrive at the sensor, i.e., in $t_{i}$. Knowing the anchor’s locations $a_{i}$ and the range-differences $\Delta r_{i}$, the sensor node performs multilateration to estimates its location as described next.

### 3.2. Multilateration Phase

We present two approaches to solve multilateration problem that will be used in the robust estimators presented in Section 6. In the first approach, we consider solving the following LS problem,
(5)$${\hat{x}}_{MLE}=\underset{x}{arg min}\sum_{i=1}^{N}{\left(\Delta \tilde{r_{i}}-(∥x-a_{0}{∥}_{2}-∥x-a_{i}{∥}_{2})\right)}^{2}$$
which is the problem (Equation 2) under i.i.d. Gaussian noise assumption. As the problem (Equation 5) is non-convex, first, we consider to study its behavior and deduce certain criteria on geometrical configurations of anchors and sensor, which helps to find the correct local minimum by an iterative method starting from a suitable initial solution. We show the contour plots of the cost function (Equation 5) in Figure 3 for different possible configurations of the anchors and a sensor (other plots are not shown due to space constraint). We observe that the cost function is “well behaved” in the following situations:
when more than three anchors (including lead) are presentwhen the sensor lies within convex-hull of the anchorswhen the anchors are placed in a symmetric fashion

By “well behaved”, we mean that the contours are almost circular and the function descends very fast near the desirable minimum, which helps an iterative solver to reach the minimum quicker when initialized near to it. By contrast, when there is a significant discrepancy with the above conditions, then the contours are elongated shape and the function descends slowly (in some of the cases almost flat) near the desirable minimum, e.g., in the top-left plot of Figure 3. From these observations, it becomes clear that a careful deployment of anchors can avoid the pitfall of the non-convex problem and an iterative solver can find the correct local minimum. We use the above points as the guidelines for the deployment of anchors.

We propose to use Gauss–Newton (GN) method (presented in Section A.1) for solving the problem (Equation 5). The GN method is computationally less expensive than methods based on convex relaxation techniques; in each iteration, it requires only *N* dot product of two vectors of size 2×1, one matrix inversion of size 2×2, and one matrix-vector multiplication of size 2×2 and 2×1. Moreover, the GN method approaches to quadratic convergence rate near the local minimum. We find that the centroid of the anchors deployed according to the above guidelines is a good initial solution and the GN method reaches close to the desired local minimum within 6 to 8 iterations for all the configurations shown in Figure 3, except in a few cases, e.g., the top-left plot where the cost function becomes very shallow near the minimum and GN method does not reach to solution in a finite number of iterations.

In the second approach, we formulate the multilateration problem in suboptimal way similar to the work in [14]. From definition of range, we write
(6)$$r_{0}^{2}={∥x-a_{0}∥}_{2}^{2}\Rightarrow x^{2}-r_{0}^{2}-2a_{0}^{T}x+a_{0}^{2}=0$$
with small misuse of the notation: $x^{T}x=x^{2}$. And from the definition of range-difference (Equation 4), we write:(7)$$\begin{array}{c} {\left(r_{0}-\Delta r_{i}\right)}^{2}=r_{i}^{2}={∥x-a_{i}∥}_{2}^{2}\;\Rightarrow \;x^{2}-a_{0}^{2}-2a_{i}^{T}x+2r_{0}\Delta r_{i}+a_{i}^{2}-\Delta r_{i}^{2}=0 \end{array}$$

By subtracting (Equation 6) from (Equation 7), we obtain:(8)$$2{\left(a_{0}-x_{i}\right)}^{T}x+2r_{0}\Delta r_{i}+x_{i}^{2}-\Delta d_{i}^{2}-a_{0}^{2}=0,\;\forall i=1,\cdots ,N.$$

This system of equations can be expressed in compact form as
(9)$$Mx+r_{0}c+d=0$$
where unknowns are $x={\left[x,y\right]}^{T}$ and $r_{0}$, and known quantities are
$$\begin{array}{cc} M & =\left[\begin{array}{cc} 2(x_{0}-x_{1}) & 2(y_{0}-y_{1})\\ \vdots & \vdots \\ 2(x_{0}-x_{N}) & 2(y_{0}-y_{N}) \end{array}\right]\in R^{N\times 2},\;c=\left[\begin{array}{c} 2\Delta r_{1}\\ \vdots \\ 2\Delta r_{N} \end{array}\right]\in R^{N}\\ d & =\left[\begin{array}{c} x_{1}^{2}+y_{1}^{2}+z_{1}^{2}-\Delta r_{1}^{2}-x_{0}^{2}-y_{0}^{2}-z_{0}^{2}+2\left(z_{0}-z_{1}\right)z\\ \vdots \\ x_{N}^{2}+y_{N}^{2}+z_{N}^{2}-\Delta r_{N}^{2}-x_{0}^{2}-y_{0}^{2}-z_{0}^{2}+2\left(z_{0}-z_{N}\right)z \end{array}\right]\in R^{N} \end{array}$$

We solve problem (Equation 9) by a method presented in Section A.2 referred to as Closed-Form (CF) method. Under low-level noise conditions, the CF estimates are close to the correct solutions, whereas under high noise conditions, they can be far from the correct solutions. In certain cases, the CF method does not produce any feasible solution or gives two ambiguous solutions. Though suboptimal, the main advantage of the CF method is its low computational cost compared to the GN method; it requires only three matrix-vector multiplications of size (2×N) and (N×1), and one matrix inversion of size 2×2. The estimate by the CF method can serve as a good initial solution for the GN method.

### 3.3. Comparing CF and GN Methods

Here, we compare the performance of the two approaches for the multilateration problem. This comparison help in choosing between the two methods for using them in the robust location estimators presented in Section 6. For simplicity, we will refer to these approaches by the name of their respective solvers. For the comparison, we consider the following two performance metrics:*mean-L2-norm-error* defined as: $E\left[∥x-\hat{x}{∥}_{2}\right]$, *std-L2-norm-error* defined as: $\sqrt{\mathrm{Var}\left[∥x-\hat{x}{∥}_{2}\right]}$,
where x and $\hat{x}$ are the true and estimated locations, respectively. Moreover, to measure the collective performance of multiple estimated locations over a ROI, we use mean of *mean-L2-norm-error* and mean of *std-L2-norm-error* over the ROI. With a slight abuse of the terminology, we refer to these two quantities as *bias* and *variance*, respectively.

We present three different simulation results comparing the performance of the two methods:In Figure 4, plots (**a**–**e**) show the localizable regions by the two methods without noise in $t_{i}$ measurements and plots (**e**–**i**) show the *mean-L2-norm-error* map with i.i.d. Gaussian noise in $t_{i}$ measurements. These plots clearly show that symmetric deployment of anchors is essential to have large localizable area in the ROI. Moreover, the GN method results into larger localizable area with lower location errors than the CF method for the same number of anchors taking part in $t_{i}$ measurements.In Figure 5, the plots show the *mean-L2-norm-error* map with i.i.d. Gaussian noise in $t_{i}$ and when the ROI is almost contained within the convex hull of the anchors. These assert that the GN method produces overall lower location errors than CF method.Table 1 shows effect of varying noise level and number of anchors on the two metrics: *bias* and *variance* by the two methods over the same settings as in Figure 5. Again, clearly the GN method results into lower *bias* and *variance* then the CF method.

We observe the GN method performs better than the CF method in all the cases. However, the CF method is significantly less expensive than the GN method. One of the two methods can be selected based on the trade-off between accuracy and computational expenses. We use these two methods alternatively in the robust location estimators to keep the overall computational costs low while still achieving sufficiently robust location estimates.

## 4. Underwater Acoustic Channel Modeling

In Section 1, we discussed the behaviors of acoustic propagation in seawater and pointed out that multipath propagation has an adverse effect on range/range-difference measurements. Several stochastic channel fading models, as discussed in Section 1.1, are proposed based on the experimental acoustic data collected at different geographical locations and seasons. As collecting a huge amount of acoustic data from the ocean environment is a costly task, few researchers [49,50] have fed information from sparsely collected experimental data into Bellhop ray tracing model and compared the simulation results with their experimentally evaluated statistical models. The results obtained through Bellhop simulations showed statistical properties similar to those obtained experimentally. These studies show that Bellhop ray tracing can be used to estimate fairly accurate channel behaviors if the parameters fed match the actual environment.

Following the idea above, we performed 2D Bellhop ray tracing simulation of a shallow sea and a moderately deep sea environment to get an overview of multipath arrivals. In Section 8.1, we provide the details of the simulation environment and the parameters. Figure A1 and Figure A2 show the simulation results for a shallow and moderately deep sea, respectively. The results show that there are several distinct paths, referred to as *eigenpaths*, over which a signal propagates from a transmitter to the receivers. Each *eigenpath* contains a stable dominant component and multiple low power randomly scattered components, collectively referred to as *eigenrays*. Each *eigenpath* can be considered as cluster of *eigenrays*. The concept of *eigenpath* and *eigenray* was first introduced in [26]. Later, it was elaborately presented in [27], which, based on experimental results and analytical considerations, suggests that the envelope of the *eigenpaths* follow a Rician fading model, and the *eigenrays* arrival angles, the amplitudes, and the phase fluctuations are independent of each other. We also find similar observations from our Bellhop ray tracing simulations as shown in the zoomed views in Figure A1b and Figure A2b.

Based on the above observations and suggestions in [14], we believe that the modified UWB S-V model [32,33] is fairly suitable for characterizing acoustic channel fading with the following supporting arguments: (i) commonly used underwater acoustic signals are, in fact, UWB signals, although their bandwidths are narrow but not negligible compared to their central frequencies; (ii) the modified UWB S-V model is based on the observations that multipath contributions generated by the same signal pulse arrive at the receiver grouped into clusters, which is similar to the observed *eigenpaths/eigenrays* behavior in the underwater channel; and (iii) as observed in [26], two Poisson models are involved in the modeling of the path arrivals in UWB communications: the first Poisson model for the first *eigenray* of each *eigenpath*, and the second Poisson model for *eigenrays* within each *eigenpath*. However, certain aspects, such as frequency selective fading, Doppler effect, and others described in [51], cannot be correctly modeled with the modified UWB S-V model. Nevertheless, the modified UWB S-V model provides a plausible statistical model for the arrival times of *eigenpaths/eigenrays* at a receiver, which is useful in quantifying the errors in range/range-difference measurements.

When the modified UWB S-V model is applied to underwater acoustic channels, the arrival of *eigenpaths* is modeled as a Poisson process with rate Λ, and within each *eigenpath*, the arrival of *eigenrays* is also be modeled as a Poisson process with the rate λ. To formally describe the arrival processes, let us define following quantities: (i) $T_{l}$ be the arrival time of the first path of the *l*th *eigenpath*, (ii) $\tau_{k,l}$ be the delay of the *k*th path within the *l*th *eigenpath* relative to the first path arrival time $T_{l}$, (iii) Λ be the *eigenpath* arrival rate, and (iv) λ be the *eigenray* arrival rate. By definition, we have $\tau_{0,l}=T_{l}$. The distribution of the *eigenpath* arrival time and the *eigenray* arrival time can be written as
(10)$$\begin{array}{cc} p(T_{l}|T_{l-1}) & =\Lambda \exp(-\Lambda \left(T_{l}-T_{l-1}\right)),\;l>0\\ p(\tau_{k,l}|\tau_{(k-1),l}) & =\lambda \exp(-\lambda \left(\tau_{k,l}-\tau_{(k-1),l}\right)),\;k>0. \end{array}$$

As discussed above, the underwater acoustic channels do not follow Rayleigh distributions, but it is appropriate to model them by Rician distribution. In the UWS S-V model, the gain of the *k*th *eigenray* within the *l*th *eigenpath* is a complex random value with a modulus $\beta_{kl}$ and phase $\theta_{kl}$. We assume that $\beta_{kl}$ value in an underwater acoustic channel are statistically independent and are Rician-distributed positive random variable, whereas the $\theta_{kl}$ values to be statistically independent uniform random variables over [0,2π]. Thus, we have $\bar{\beta_{kl}^{2}}=\bar{\beta_{00}^{2}}\exp\left(-T_{l}/\Gamma \right)\exp\left(\tau_{k,l}/\gamma \right)$ where $\bar{\beta_{00}}$ represents the average energy of the first *eigenray* of the first *eigenpath*, whereas Γ and γ are the power decay coefficient for *eigenpath* and *eigenrays* within it, respectively. According to the above equation, the average power decay profile is characterized by an exponential decay of the *eigenpath* and a different exponential decay for the amplitude of the *eigenrays* within each *eigenpath*. Note that in some cases the *eigenpaths* may overlap each other. For example, if for some *k*, $\tau_{k,l}\geq T_{l+1}-T_{l}$, the *l*th and (l+1)th clusters overlap for all subsequent values of *k*.

## 5. Modeling Errors in Range-Difference Measurements

In Section 1.1, we pointed out the sources of errors in range-difference measurements are (i) the delay in the receiver system, (ii) variable sound speed, and (iii) multipath propagation. Accurate modeling of the uncertainties in measurements due to these factors would require specific information about the environment, and thus it is a difficult task. Here, we model them approximately based on empirical observations with possible theoretical justifications.

Errors in range measurements due to delay in the receiver system are dependent upon the electronics of the receiver. We can safely assume that the errors follow zero-mean i.i.d. Gaussian distribution with small variance, which is the case in many commercially available acoustic modems.

Sound speed in the ocean depends on the temperature, salinity, and pressure. These factors vary randomly both temporally and spatially from small to large scale. Modeling the overall variations of sound speed over a ROI and then calculating range-difference is a tedious task. Rather, we take a simple approach and assume a constant sound speed, typically an average of few measured samples throughout the ROI. This can result in both positive and negative errors in the range measurements. Further, we assume that when an acoustic signal travels a short distance then it experiences a few small variations of speed, and when it travels a long distance it can experience several large variations. In the latter case, the positive and negative variations can nullify each other’s effects in many situations resulting in small errors, but, in some cases, there can be only large positive or negative variations resulting in large errors. Thus, we can model the range errors from multiple transmitters at different distances to follow approximately an i.i.d. Laplacian distribution with unknown parameters dependent on the distances. For the sake of mathematical simplicity, we approximate the errors in range-difference measurements to follow an i.i.d. Laplacian distribution (The difference between two random variables following i.i.d. Laplacian distributions can be modeled correctly by a Generalized Normal distribution, an intermediate between Laplacian and Gaussian distribution.).

As discussed in Section 4, the multipath spreading causes the problem of resolving the first arriving *eigenray*. Thanks to the advances in the digital modulation–demodulation technique such as direct-sequence spread-spectrum (DSSS) binary phase-shift keying (BPSK), the receivers can separate the multipaths via the despreading operation that suppresses the time-spreading induced interference [16,17,52,53]. In many situations, the receiver can distinguish the first *eigenray* of the first *eigenpath*, except when it receives many closely packed *eigenrays*, which often happens in shallow water. From the Bellhop ray tracing simulation, we have the following observations:In shallow sea, numerous *eigenpaths* and *eigenrays* arrive at the receivers with significant strength and their strength decays slowly, whereas in deep sea, only few *eigenpath* and *eigenrays* arrive at the receivers with significant strength and their strength decay fast.Densely packed *eigenrays* in a *eigenpath* are generally non-resolvable.The vertical links exhibit narrower multipath spreading, while slant and horizontal links exhibit wider multipath spreading ranging from a few to hundreds of milliseconds; larger the distance between transmitter and receiver, higher are the chances of wider spreading.In the case of slant and horizontal links, often there is no direct ray arriving at the receiver distant from a transmitter.

From these observations, we can conclude that range measurements due to multipath propagation have only positive errors and are due to the following two reasons: (i) assuming that a receiver can detect the first *eigenpath*, there are high chances of non-resolvability among *eigenrays* within it, and (ii) there are chances that no direct rays arrive at the receiver. The errors due to the first reason are much frequent and smaller and due to the second reason are less frequent and larger. Thus, the overall errors in range measurements can be approximated by an i.i.d. Exponential distribution. This approximation is per the fact that time intervals among the arrival of *eigenrays* are modeled by exponential distribution as described by the modified UWB S-V model presented in Section 4. Considering a range-difference as a pairwise difference between two ranges, the errors in it can be modeled by an i.i.d. Laplacian distribution.

Combining the effects of receiver electronics, variable sound speed and multipath propagation on range-difference measurements, we approximate the errors to follow an i.i.d. Laplacian distribution centered around zero with unknown scale parameter. This approximation may not always capture correctly the joint effect, but in many situations it is sufficient and mathematically simple. If $t_{i}$ represent the ToA at a receiver from *i*th anchors, then we write the TDoA with respect to 0th anchor as: $\Delta t_{i}=t_{0}-t_{i}$ and the measured TDoA as $\Delta {\tilde{t}}_{i}=\Delta t_{i}+\varepsilon_{i}$ where $\varepsilon_{i}∼\mathrm{Laplace}\left(0,b\right)$ represent the error following i.i.d. Laplacian distribution. Thus, the measured range-differences can be written as
(11)$$\Delta {\tilde{r}}_{i}=\bar{\nu }\Delta {\tilde{t}}_{i}+\epsilon_{i},\;\forall i=1,\cdots ,N$$
where $\bar{\nu }$ represents an average sound speed evaluated over the ROI and $\epsilon_{i}$ represent the errors following an i.i.d. Laplacian distribution.

## 6. Robust Location Estimators

As discussed in Section 5, the errors in range-difference measurements follow an i.i.d. Laplacian distribution, which accounts for both the frequently occurring small errors and rarely occurring large errors. Without any further information about the varying sound speed and the multipath propagation, extreme values in range- difference measurements can be considered as outliers. The presence of outliers in the measurements has a significant adverse effect on the estimated locations. Thus, robust estimators, which can identify and avoid outliers during the estimation process, are essential for the localization of UWASNs.

Robust estimators can be evaluated and compared using different criteria. For our application, the *breakdown point*, and *mean-L2-norm-error* are of particular interest. The *breakdown point* is defined as the largest proportion of the outliers that the measurements may contain such that the resulting estimate by an estimator remains close to the true parameter. The smallest possible breakdown point is 1/N, which tends to 0 when the sample size *N* becomes large. This is the case for an LS estimator. Another important criterion to measure robustness is *influence function*, which measures the rate at which the estimator responds to a small amount of contamination in the independent variable (i.e., in *X*-space). In our application, this refers to the errors in the anchor positions, which can be avoided by using accurate GPS attached to the anchors.

Introduction of robustness into an estimator comes at the cost of decreased *statistical efficiency*, i.e., unwanted high variance of the estimate. A robust estimator with a good trade-off between *breakdown point*, *statistical efficiency*, and computational expense is desirable. As the consequences of the Laplacian distribution of the errors in the range-difference measurements, the first robust estimator we consider in this paper is the LAD, which we already introduced in Section 2. However, as discussed in Section 5, the errors may not always strictly follow the Laplacian distribution. Therefore, we also consider two other popular robust estimators: LMedS and RANSAC, for their effectiveness and low computation expenses. In the following, we present each of these robust estimators and compare their performances according to the above criteria.

As discussed in Section 5, the range/range-difference measurements due to distant anchors, i.e., having horizontal or slant links, are highly susceptible to large errors due to multipath propagation and varying sound speed than the anchors that are closer and/or having vertical links. Taking this into account, we propose to avoid the distant anchors in the estimation as a preliminary step of the robust estimators. We do this by initially estimating the sensor location by the CF method using the range-differences measurements from all the available anchors. Of course, this initial location estimate may not be very accurate, but it is sufficient to choose a few distant anchors among the others. We include this *preprocessing* step in all the robust estimators described below.

### 6.1. LAD Estimator

The LAD estimator (Equation 3) is a MLE under i.i.d. Laplacian noise condition and was first proposed in [54]. The LAD estimator generalizes the median of a one-dimensional sample, but it has 0%
*breakdown point* same as the LS estimator, whereas the breakdown point of the sample median is 50%. This is because the LAD cannot cope with grossly aberrant values in the independent variable space, i.e., *X*-space. However, the LAD estimator is robust against a high number of outliers in the dependent variable, i.e., *Y*-space, which is range-difference measurements in our case. As the LAD estimator is an MLE, thus it has high *statistical efficiency*. To solve the LAD problem (Equation 3), we choose the Iterative Reweighted Least-Squares (IRLS) method [55] owing to its ease of use and good performance. IRLS solves the LAD problem (Equation 3) by iteratively solving the following weighted-LS problem:$$x^{\left(t+1\right)}=\underset{x}{arg min}\;\sum_{i=1}^{N}\omega_{i}^{\left(t\right)}\;e_{i}^{2}\left(x\right),$$
where $e_{i}\left(x\right)=\Delta {\tilde{r}}_{i}-(∥x-a_{0}{∥}_{2}-∥x-a_{i}{∥}_{2})$ referred to as residue function. The weights is initially chosen as $\omega_{i}^{\left(0\right)}=1$ and later as $\omega_{i}^{\left(t\right)}={\left|e\left(x^{\left(t\right)}\right)\right|}^{-1}$. IRLS can converge to good local minimum if it starts with an initial solution close enough to it. The advantage of the IRLS method is its low computational requirement; its iterations can be efficiently and accurately solved by the GN method described in Section A.1. The iterations of IRLS can be stopped when the change between the two consecutive estimates is sufficiently small, i.e., $∥x^{\left(t+1\right)}-x^{\left(t\right)}{∥}_{2}\;\leq 0.0001$. With this stopping criterion, we find that 30–40 iterations are sufficient to reach acceptable solution in all our simulations where in each iteration the GN method takes around 4–6 iterations. The LAD estimator tailored to robust localization is presented in Algorithm 1. Note that in Line 8, η>0 is used for regularization to avoid division by zero and should be sufficiently small, e.g., 0.0001. This has similar effect as Huber regularization [56].
**Algorithm 1:** Least-Absolute-Squares (LAD) Robust Location Estimator
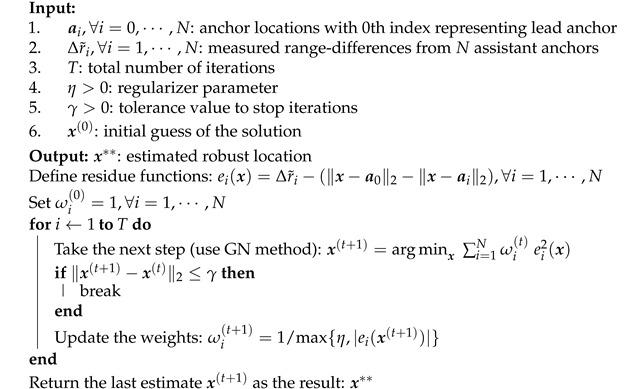


### 6.2. LMedS Estimator

The efforts to develop a robust estimator with a high breakdown point gave birth to more generalized versions of LAD estimator such as M-estimator and others described in [56]. However, they could not achieve a high breakdown point in simple regression. In this direction, a method called *repeated median* was proposed in [57], which can achieve a breakdown point up to 50%. However, this method is not affine equivariant and computationally expensive (increase exponentially with the dimension of the parameter to be estimated). The estimator proposed in [58] replaces the sum in the LS method by the median, thus called LMedS, achieves *breakdown point* up to 50%. It is given by
$$\underset{x}{arg min}\;\underset{i}{\mathrm{median}}\;e_{i}^{2}\left(x\right),\;\forall i=1,\cdots ,N.$$
where $e_{i}\left(x\right)$ is residues defined Section 6.1. However, it has a low *statistical efficiency* when measurements contain Gaussian noise [58]. LMedS estimator does not have an analytical solution. It searches in the space of solution obtained from subsets of the minimum number of measurements. We present the LMedS estimator tailored to robust localization in Algorithm 2. Note that even though it is sufficient to have subsets of 2 assistant anchors to estimate an unknown location, in Algorithm 2 we consider subsets of 3 assistant anchors because it gives a better estimate and reduces the chances of not having a feasible solution by the CF method as discussed in Section 3.
**Algorithm 2:** Least-Median-Squares (LMedS) Robust Location Estimator
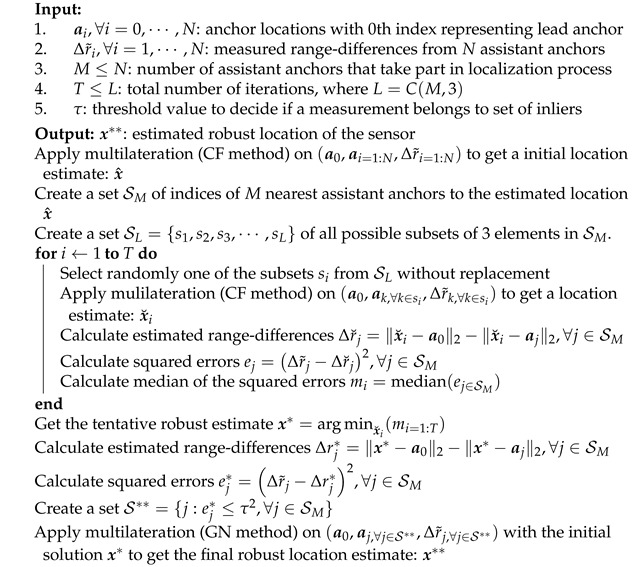


At the end of the basic LMedS iterations, the algorithm returns a robust estimate using a subset of three assistant anchors, which results in the minimum of median errors. However, it may be the case that the other assistant anchors, which are not included in the robust subset, can also be good candidates (inliers) to the current estimated location. Thus, to improve the *statistical efficiency* of LMedS, we use the basic LMedS estimator as a tool to distinguish between inlier and outlier anchors. We use a threshold value τ>0 to reject the anchors for which ${\left(\Delta {\tilde{r}}_{j}-\Delta r_{j}^{*}\right)}^{2}>\tau^{2}$ and keep the remaining anchors as final set of inliers. Selecting the value of the τ can be difficult when no further information is available about the noise in the measurements, and it is selected empirically in many practical cases. In the case, when the noise in the inlier measurements are assumed to be i.i.d. Gaussian with zero mean and standard deviation σ, then $\tau^{2}$ can be fixed equal to $3.84\;\sigma^{2}$ for 5% probability of rejecting an inlier; see [59] (p. 118) for details behind this calculation.

### 6.3. RANSAC Estimator

The RANSAC algorithm was first introduced by Fischler et al. [60], as a robust method for points matching in image analysis and automated cartography applications. RANSAC can achieve more than 50% breakdown point and is suitable for the cases when the measurements contain many outliers. The main steps of the RANSAC are similar to LMedS, i.e., hypothesize and then test. In the first step, the algorithm estimates the model parameter using a randomly selected subset of measurements, and in the second step, it checks which measurements among all are consistent with the estimated model from the first step. The set of such measurements is called *consensus set*. To do so, it uses a threshold τ>0, which is chosen empirically when no statistics is known about the noise in the non-spurious measurements; otherwise, as discussed above, $\tau^{2}$ can be fixed equal to $3.84\;\sigma^{2}$ for 5% probability of rejecting an inlier. Another input it requires is the cardinality *K* of the consensus set to decide if it is big enough to be considered as a robust set. The value of *K* is again chosen empirically; otherwise, if we have an idea about the probability of outlier measurements i.e., $p_{0}$, then $K=(1-p_{0})N$ is selected. A major difference between the LMedS and the RANSAC is the way the estimated models are scored. In the former, it is scored by the median of the distances to all the measurements, and the one with the least median is selected. In the latter, it is scored by a threshold value and the one with the largest cardinality of the consensus set is chosen. The RANSAC has been very successful for robust estimation, but one of the problems with it is that if the threshold τ is set too high then the robust estimate can be very poor. This is because the RANSAC in effect finds the minimum of a cost function:$$C_{1}=\sum_{i}\rho \left(e_{i}^{2}\right),\;\mathrm{where}\;\rho \left(e^{2}\right)=\left\{\begin{array}{cc} 0, & \mathrm{if}\;e^{2}<\tau^{2}\\ \mathrm{contant}, & \mathrm{otherwise} \end{array}\right.$$
which implies that the inliers score nothing, whereas each outlier scores a constant penalty. Thus, the higher $\tau^{2}$, the more solutions have equal values of $C_{1}$, tending to poor estimations. In Algorithm 3, we present the improvised version of the RANSAC proposed in [61], which is referred to as *m-estimator sample consensus* (MSAC). In MSAC, rather than minimizing cost function $C_{1}$, a new cost function is minimized:$$C_{2}=\sum_{i}\zeta \left(e_{i}^{2}\right),\mathrm{where}\;\zeta \left(e^{2}\right)=\left\{\begin{array}{cc} e^{2}, & \mathrm{if}\;e^{2}<\tau^{2}\\ \tau^{2}, & \mathrm{otherwise} \end{array}\right.$$
which implies that outliers are still given a fixed penalty but, now inliers are scored on how well they fit the measurements. This improvisation always yields a better estimate than the original RANSAC with no extra computational expense. In Algorithm 3, line 13 calculates this new cost function $C_{2}$ and the algorithm finally select robust estimate which yields the lowest cost. In this paper, even if we referred to Algorithm 3 as RANSAC, but in the implementation, we use the improved version, the MSAC.

In both the LMedS and the RANSAC algorithms, an exhaustive search over all the possible subsets of the 3 assistant anchors is not necessary to find the robust estimate. In fact, if we have an idea about the proportion of the outlier measurements, i.e., $p_{o}$ the probability that a measurement is an outlier, and if we fix the desired probability of success, i.e., $p_{s}$ the probability that we get at least one good subset without any outlier, then it is sufficient to search over approximately:$$T=\frac{\log(1-p_{s})}{\log\left(1-{\left(1-p_{o}\right)}^{3}\right)}$$
random subsets to find the robust estimate, see in [59] (p. 119) for details behind this calculation. For example, if there are 30% outliers in the measurements, then we can terminate the algorithm after 11 iterations to have 99% chance of getting a subset of anchors with good measurements. In practice, its always better to search over a little larger number of subsets than the above prediction.

Note that in both the LMedS and RANSAC algorithms, we utilize the CF method while searching for the robust subset and then use the GN method for the final robust estimate. With this trick, we reduce the overall computational expenses of both the algorithms, but without significantly sacrificing the robustness of their final estimates. Among the two algorithms, the RANSAC is slightly more expensive and generally produces a better estimate than the LMedS.
**Algorithm 3:** RANSAC Robust Location Estimator
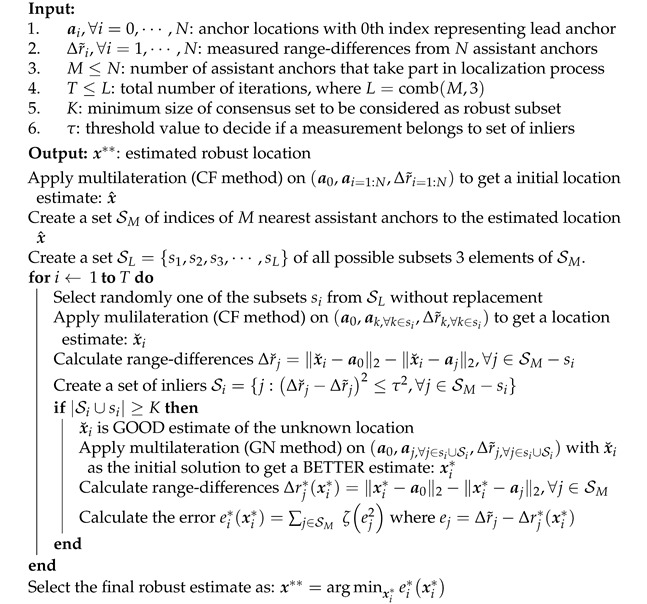


## 7. Localization Using Surface Mobile Anchor(s)

In Section 6, we saw that the higher the number of range-difference measurements, the higher the chance that a robust estimator produces an accurate and reliable location estimate. However, deploying and maintaining a large number of fixed anchors is not practical for long-term applications. Anchors deployed deep into the water have the problem of maintaining their batteries and accurate locations, while anchors deployed on the surface do not face these issues, but have higher chances of being disturbed (carried away) by surface vehicles such as ships and fishing boats. Using surface mobile anchor(s) can easily avoid these problems, especially for localizing large-scale networks, while providing other benefits. A small low-cost surface vehicle with (i) a precise clock, (ii) an accurate GPS, (iii) an acoustic modem, (iv) a long-range radio communication, (v) autopilot navigation, and (vi) a power supply to sustain a few hours of mission can serve as a suitable mobile anchor. For example, the Evologics Sonobot system (https://evologics.de/sonobot-system) is one of the good candidates as a mobile anchor. Once the sensors are deployed in a ROI, then a single or multiple surface mobile anchors can navigate over the ROI and broadcast beacon signals to locate the sensors below the surface. Depending upon the scenario, e.g., turbulence in the ROI, this can be repeated periodically, e.g., every day or every week, to localize the sensors.

To implement UPS using a mobile anchor, we need to imitate the range-difference measurement phase of the UPS described in Section 3. More generally, depending on the resources available at hand, we can perform it by either a single mobile anchor or a combination of a fixed and a single or multiple mobile anchors to be able to scale with the size of the network or finish the localization process quicker. In all these cases, a mobile anchor acts as a virtual stationary anchor at different instants of time. A common requirement of localization schemes using mobile anchors is to maximize *coverage*, i.e., number of the sensors localized robustly in the ROI, with possibly minimum movements of the mobile anchor(s). Another important aspect is dealing with the loss of acoustic signals due to environmental conditions (noise, interference, obstacle, and absorption), which often happens in shallow seas. The lead anchor’s beacon signal is of prime importance because all the TDoAs at the sensors are calculated based on its arrival. Thus, in our proposed schemes presented below, we keep a provision of repeating its beacon signals at constant intervals while the mobile anchor(s) moves around in ROI. In Section 3, we showed that symmetric placement of the anchors in the ROI achieves maximum *coverage* as depicted in Figure 4. With the assumption that all the anchors are omnidirectional and have the same acoustic range, a star topology with the lead anchor at the center and the assistant anchors on a circle, as shown in Figure 6, is an optimal placement to achieve maximum *coverage*. Below we present three different configurations of a fixed and/or multiple mobile anchors to perform robust silent localization of mid-scale to large-scale networks. When a mobile anchor completes moving over a circle, we refer it to as a *beacon cycle* and the parts of the cycle between the two consequent beacon signals by the lead anchor as a *beacon segment*. To avoid errors introduced due to motion, mobile anchor stops for a short duration when it broadcasts its beacon signals.

### 7.1. A Stationary and a Mobile Anchor

This configuration is suitable for the situation when there is already a fixed central gateway node deployed in the ROI that can also act as the lead anchor. This is a commonly used network topology to collect data from the sensors, e.g., using TDMA protocol [6]. Usually, the gateway node lies on the surface in the middle of the ROI with an acoustic modem hanging below the surface and a long-range radio to communicate with a base station at the shore. In this configuration, a mobile anchor moves on an approximately circular path at the periphery of the ROI while broadcasting beacon signals from different locations as shown in Figure 6a. The process starts with the broadcast of the beacon signal from the lead anchor represented as **A0** at Position 0. The beacon signal contains an indicator for the start of a *beacon cycle* and the GPS location of **A0**. At this instant, the mobile anchor, represented as **A1**, is at Position 1. As soon as **A1** receives the beacon signal from **A0**, it broadcasts its beacon signal containing its current location and the delay between the current time and the instant it received the beacon signal from **A0**. Now, the mobile anchor moves to Position 2 represented as **A2**. Here, it calculates the time instant the beacons signal from **A0** would have arrived at **A2** knowing the distance between Position 0 and 2 and a constant sound speed $\bar{v}$. In fact, if the mobile anchor was moving exactly on a circle, then the beacon signal from **A0** would reach at the same instant to all the future positions of the assistant anchor. **A2** broadcasts its beacon signal containing its current location and delay between the current time and the instant **A2** received the beacon signal from **A0**. Next, the mobile anchor moves to Position 3 represented as **A3** and broadcasts its beacon signal containing similar information. This completes the first *beacon segment* of the *beacon cycle*. Now, the mobile anchor moves to Position 4 represented as **A1** again. A new *beacon segment* starts with the broadcast of the beacon signal from **A0** and then the mobile anchor keeps moving forwards while broadcasting its beacon signal as it did in the last *beacon segment*. The mobile anchor broadcasts its last beacon signal from Position 12 represented as **A3**. Finally, the lead anchor broadcasts a signal indicating the end of *beacon cycle* and this completes the beacon signaling process.

### 7.2. A Single Mobile Anchor

In this configuration, a single mobile anchor serves both as the lead anchor and the assistant anchors at different instants of time. This configuration is suitable when even a single permanent node cannot be deployed due to certain difficulties. In this situation, the mobile anchor can also serve as the gateway node to harvest data from the sensors. This configuration is easily scalable and can localize a large-scale network beyond the acoustic range of a central anchor. In such a situation, the whole ROI is divided into multiple cells, and a mobile anchor localizes each cell sequentially. Figure 6b depicts the navigation plan of a mobile anchor for a single cell. The process starts when the mobile anchor is at Position 0 acting as the lead anchor represented as **A0** and broadcasts an indicator signal for the start of *beacon cycle* with its current location. The mobile anchor then moves to Position 1 and acts as assistant anchor represented as **A1**. Here, it broadcasts a beacons signal containing its current location, and the delay between the current time and the instant **A1** would have received beacon signal from **A0**. The mobile anchor now moves to the Position 2 and then to the Position 3 while broadcasting the beacon signals containing its locations, and the delay information calculated as earlier. Now it moves back to the Position 0, and this completes the first segment of the *beacon cycle*. Here, again, it acts as the lead anchor represented as **A0** and broadcasts its beacon signal containing its current location and an indicator for a new *beacon segment*. Then, the mobile anchor continues moving forward and broadcasting beacon signals in the same pattern as it did in the last segment until it finally reaches the Position 0 after broadcasting the last beacon signal at Position 12 as being anchor **A3**. At Position 0, it broadcasts the signal indicating the end of *beacon cycle*, which completes the beacon signaling process of the current cell. Now, the mobile anchor moves to the adjacent cell and repeats the same signaling pattern, and continue the process until it covers all the cells in the ROI.

### 7.3. Two Mobile Anchors and More

Similar to the single mobile anchor configuration, this is suitable for the situation when no permanent anchor can be deployed and a large-scale sensor network must be localized. Using two or more mobile anchors simultaneously can speed up the whole process of localizing a large-scale network. Unlike the single mobile anchor, this configuration may require a sophisticated navigation plan and signal broadcasting schemes among the multiple mobile anchors. Without going into details of optimal path planning and signal scheduling schemes, here we present two simple strategies of using two mobile anchors to localize a large-scale sensor network while keeping a balance between the total time and the energy consumed by the two mobile anchors. As discussed above, the whole ROI can be divided into multiple cells and then localized by the two mobile anchors. In the first strategy, the two mobile anchors together localize a single cell such that one stays at the center of the cell acting as the lead anchor while the other moves on the periphery of the cell acting as assistant anchors as shown in Figure 6a. Once they complete the beacon signaling of one cell, then they move to the next cell and exchange their roles. The two mobile anchors continue in this pattern until they finish signaling all the cells. In the second strategy, the two mobile anchors simultaneously and independently localize two distant cells avoiding acoustic interference between them and moving in the pattern shown in Figure 6b. The process continues until they finish the beacon signaling all the cells.

### 7.4. Parameter Selection: Location Accuracy and Overall Cost Trade-Off

After a *beacon cycle* in a cell is completed, the sensors within that cell can estimate their locations using one of the robust estimator presented in Section 6. The robustness of the estimated location of a sensor depends upon the number of received beacon signals that are less affected by multipath spreading and variable sound speed. This consequently depends upon the environmental conditions in the ROI, which cannot be avoided but can be mitigated up to a certain extent by investing a higher number of beacon signals and higher computational cost of the sensor nodes. In the environment where the chances of acoustic packet losses are higher, the configurations presented in the preceding sections need to incorporate a higher number of assistant anchors in a *beacon cycle*. To ensure that the sensors can use the robust estimators in Section 6, a *beacon cycle* should be divided into sufficient number of *beacon segments*. In the environment where the chances of multipath spreading are higher and increase with the distance, the anchors and the sensors should be kept close, i.e., small diameter of a circular cell. This will effectively increase the chances of receiving a higher number of beacon signals with the least multipath spreading, and therefore of being favorable for robust location estimate. The diameter *d* of the circular path of the mobile anchor is bounded below by the acoustic range *r* of the anchors and also depends upon the depth *h* of the deepest sensor to be localized. It is geometrically given as $d=\sqrt{r^{2}-h^{2}}$, where h<r should be satisfied to ensure that the sensors receive beacon signals from multiple anchors as shown in the Figure 6c. Keeping a small value of *d* also ensures more vertical links between the anchors and the sensors, i.e., small θ, which results in reduced multipath spreading. However, this also requires to divide the whole ROI into more number of smaller cells to cover it and requiring a longer time to localize it. In a deep sea scenario, it is possible to relax this criterion up to a certain extent, i.e., keep θ sufficiently large, without sacrificing significantly the robustness of estimated locations, but in the shallow sea where the multipath propagation is prominent, we must keep θ small. Moreover, depending upon the size of *d*, the cells need to overlap among each other to cover the gaps created at the boundaries of the cells as shown in Figure 6d. The sensors in the overlapping areas can receive more number of beacon signals from the *beacon cycles* in the adjacent cells than the sensors in the central region of the cells. These sensors can utilize this extra information to improve their location estimate, e.g., averaging location estimates due adjacent *beacon cycles*, which otherwise were less accurate, as shown in Figure 4f–i.

Increasing the number of assistant anchors to deal with possible loss of acoustic signals, decreasing the diameter of the circular path of mobile anchors to mitigate the effect of multipath spreading, or increasing the overlap among the cells, enhances the robustness of estimated locations of the sensors. However, all these come at the cost of high computational requirements of the sensor nodes, i.e., large memory, and computations to process the high number of beacon signals, as well as the high expenses (e.g., long time) incurred at the mobile anchors. Thus, a trade-off between the two aspects should be maintained, and it highly depends upon the application scenarios. The three mobile anchors schemes presented in the preceding sections do not have performance advantages over each other in term of the robustness of the estimated locations but are possible practical schemes for localizing network that can easily scale with the size of the network and the availability of the mobile anchors, i.e., more the number of mobile anchors, the quicker the localization of a large ROI finishes. The trade-off between the robustness of the estimated locations and overall cost can be better evaluated only by conducting real sea trials.

## 8. Simulations and Results

We performed BELLHOP ray tracing simulation to study the multipath propagation in particular scenarios and numerous simulations to evaluate the performance of the robust estimators presented in Section 6. Below, we describe in detail the simulations and their results.

### 8.1. BELLHOP Ray Tracing to Study Acoustic Multipath Propagation

We perform 2D BELLHOP ray tracing simulations of a shallow and a moderately deep sea environment with varying sound speed profile (SSP) and non-flat boundaries conditions. In the simulations, we use a signal of frequency 28 kHz typically used for an acoustic range between 4 and 6 km and use the UNESCO equation to calculate the SSP by feeding a typical salinity and temperature profile. We use a randomly varying seabed profile and a sinusoidal surface of 1 m amplitude considering relatively calm waves. In both the cases, we place a transmitter at a depth of 5 m below the surface and 6 receivers at different ranges from the transmitter. In the case of shallow sea environment, the receivers are at depth of 40 m, and the seabed is at depth of 75 m, while in the case of the moderately deep sea environment, the receivers are at the depth of 200 m, and the seabed was at the depth of 230 m. The *eigenrays* plots along with the SSP plots and the amplitude vs arrival times plots for the different receivers are shown in Figure A1 and Figure A2. From the simulation results, we deduced certain observations presented in Section 4, which support our choice of using the modified UWB S-V model for the multipath propagation in underwater acoustics.

### 8.2. Comparing the Robust Location Estimators

We compare the performances of the three robust estimators based on numerous Monte Carlo simulations under different noise conditions in the ToA ($t_{i}$) measurements. For comparison, we use the two metrics: *bias* and *variance*, defined in Section 3. To include the effects of varying sound speed and multipath propagation, we deliberately corrupt some of the ToA measurements by adding relatively large random values generated from i.i.d. Uniform distributions. In all of the simulation trials, we have 121 sensors deployed over a rectangular grid of size 11×11 covering an area of 4000×4000 m${}^{2}$, which are localized using 13 anchors as shown in Figure 5e. The anchors lie on the water surface, whereas the sensors are at depth of 100 m. We assume all the sensor lie within the acoustic range of all the anchors (i.e., acoustic range is greater than 5 km) and the average sound speed is $\bar{v}=1530$ m/s. As the *preprocessing step*, suggested in Section 6, the LMedS and RANSAC filters out the two farthest among the 12 assistant anchors, i.e., effectively N=12 and M=10. In the following, we describe the different simulation cases and present their results.

#### 8.2.1. Gaussian Noise

First, we consider comparing the estimators in the well-studied Gaussian noise condition. Therefore, we add zero-mean i.i.d. Gaussian noise in the ToA measurements with two different values of standard deviation σ=1.0 and σ=2.0 ms, respectively. To create outliers in the measurements, we add random values from i.i.d. Uniform distributions from interval is [−30,−10]⋃[10,30] ms. The threshold values are chosen empirically to achieve low *bias* and is set to $\tau^{2}=16.0\;{\left(\sigma \;\bar{v}\right)}^{2}$ m${}^{2}$ in both the LMedS and the RANSAC estimators. Note that in the simulations, we do not directly introduce i.i.d. Gaussian noise in the range-difference measurements, but instead into the three ToAs measurements as it happens in the UPS. Thus, we do not use the formula suggested in LMedS and RANSAC for setting the value of τ. We select the parameter *K* in the RANSAC as K=10−q, where *q* is the number of outliers. To avoid the effect of the random search on the performance of the LMedS and RANSAC, we set the number of iterations T=120, i.e., they searched over all possible subsets of three anchors among 10 assistant anchors. The bar plots in the second row show results for the case with T=35.

From the simulation results shown as the bar plots in Figure 7, we have the following observations and the possible explanations for them:In the case of no outliers, the RANSAC performed similar to the GN method, whereas the LAD performed slightly lower than the GN method and the LMedS performs slightly lower than all of the estimators in term of both the metrics (*bias* and *variance*).-The GN method solves an MLE under Gaussian noise condition, thus it performs better than others. The improvisation in the RANSAC makes it behave like an MLE under Gaussian noise in absence of outliers. The LAD and the LMedS estimators are not optimal under Gaussian noise condition.In the presence of outliers and low-level noise in the inlier measurements (1st row of Figure 7), the LMedS and RANSAC performed similar to each other, whereas the LAD performed lower than the two in term of both the metrics.-The LMedS and the RANSAC exclusively filter out the outliers and use the LS for their final estimates, whereas the LAD does not exclusively filter out the outliers but weighs them according to their errors from the best fit; thus, some reminiscent of the outliers affect the final estimate. Moreover, the LAD is not optimal estimator under Gaussian noise condition.In the presence of outliers and high-level noise in the inlier measurements (3rd row of Figure 7), the RANSAC performs slightly better than the others in term of the *bias*. The LAD and the LMedS performed similarly to each other in the term of the *bias* but the LAD performed better than the other two in the term of the *variance*, specifically when there are a high number of outliers.-Again, the LAD and the LMedS are not optimal estimators under Gaussian noise condition while the improvisation in the RANSAC makes it behave like an MLE under Gaussian noise condition on the selected inliers. The GN and the LAD are solving MLE, thus their *variance* should be lower than non-MLE estimators.In the case of low-level noise in the inlier measurements (1st row of Figure 7), a noticeable *breakdown* happened for LMedS when the number of outliers is 6 (i.e., more than 50%), while the LAD and RANSAC were on the verge of *breakdown*. However, in the case of high-level noise (3rd row of Figure 7), the *breakdown* happened for all the three robust estimators, but the LAD still performed marginally better than the other two robust estimators.-Theoretically, the LMedS can handle up to 50% outliers, whereas the RANSAC can cope with more than 50% of outliers in a large set of data points. Similarly, the LAD can also cope with a high number of outliers in *Y*-space. In presence of high noise in the inlier measurements, there are high chances that one of the inliers out of the four can lie far away from others; thus, the RANSAC leaves the inlier and instead selects the outlier in the final estimate. The LAD involves all the available measurements but weighs them according to their errors from the best fit model at each iteration. Thus, this ensures that the LAD does not perform worse than the GN method if the initial solution was selected close enough to the global solution.The third row in Figure 7 show that the LMedS and RANSAC can find the robust estimates without searching over all the possible subsets, e.g., 35 iterations are sufficient to have 99% chance of getting a good subset when 50% measurements are outliers.

Overall, the RANSAC performed better than all the other estimators except for a few cases. Nevertheless, the LMedS did not perform significantly lower than the RANSAC while being computationally less demanding than the other two robust estimators.

#### 8.2.2. Laplacian Noise

As discussed in Section 5, the errors in TDoA measurements can be approximately modeled by i.i.d. Laplacian distribution; thus, in this simulation, we considered i.i.d. Exponential noise in the ToA measurements. Again, we consider two levels of noise in the ToA measurement, so we choose two scales: σ=1.0 ms and σ=2.0 ms in the exponential distributions. Although an exponential distribution allows extreme values (outliers), to be discrete on the number of outliers, we deliberately added random values from i.i.d. Uniform distribution in the interval [−30,−10]⋃[10,30] ms into some of the ToA measurements. The parameters for the LMedS and the RANSAC were chosen as explained in the case of Gaussian noise simulations presented above.

From the simulation results, as shown in Figure 8, we have the following observations and the possible explanations for them:In the case of no outliers, the LAD estimator performed slightly better than all the other estimators in terms of the *bias* and almost similar to the GN method in terms of *variance*.-The LAD is an MLE under the Laplacian noise condition.In the presence of outliers and low-level noise in the inlier measurements, which are well separable from outlier measurements (1st row of Figure 8), the LMedS and the RANSAC performed similar to each other and better than the LAD in the term of both the metrics.-The LMedS and the RANSAC exclusively filter out the outliers and use the LS for the final estimate, whereas the LAD does not exclusively filter out the outliers but weighs them according to their errors from the best fit model; thus, some reminiscent of the outliers may deteriorate the final estimate.In presence of either high-level noise in the inlier measurements (2nd row of Figure 8), the LAD and the RANSAC performed similar to each other and slightly better than the LMedS in term of the *bias*, while the LAD performed better than the other two robust estimators in term of *variance*.In presence of low-level noise in the inlier measurements (1st row of Figure 8), the *breakdown* happened only for the LMedS when there were more than 50% outliers, otherwise, the *breakdown* happened for both the LMedS and the RANSAC.-As discussed above, the LMedS cannot handle more than 50% outliers. The RANSAC can cope with more than 50% outliers, but when the inlier and outlier measurements are well separable. However, in the case of high-level noise, the inliers are outliers are not well separated and the RANSAC leaves an inlier and instead selects an outlier in its final estimate. The LAD involves all the available measurements but weighs them inversely according to their errors from the best fit model at each iteration. Thus, this ensures that the LAD does not perform worse than the GN method if the initial solution was selected close enough to the global solution.

Again, the RANSAC performed better than the other two robust estimators in the case of low-level noise in the inlier measurements otherwise the LAD performed almost similar or better than the RANSAC in term of both metrics.

#### 8.2.3. Laplacian Noise with Distance Dependent Scale

As discussed in Section 5, the errors in the range measurements can be distance-dependent. Therefore, we imitated this situation by adding i.i.d. Exponential noise to the ToA measurements whose scale σ increases with the distance: σ=d/3000 ms, where *d* in meters is the actual distance between the transmitter and the receiver. Again, to be discrete on the number of outliers, we deliberately added random values from an i.i.d. Uniform distribution in the interval [−15,−5]⋃[5,15] ms to some of the ToA measurements. In this situation, there are chances that the total number of outlier measurements are not equal to the deliberately created ones. Thus, there is uncertainty about the number of outliers and the threshold value to distinguish between inlier and outlier measurements. For the LMedS and the RANSAC, the thresholds are chosen empirically as: $\tau^{2}=20{\left(\bar{v}/1000\right)}^{2}$ m${}^{2}$ and the number of iterations are set to T=120. The cardinality of the consensus set in the RANSAC is chosen as K=10−q, where *q* is the number of deliberately created outliers.

The bar plots in Figure 9 show that the LAD estimator performs slightly better than the other two robust estimators in term of both the metrics. The situation considered in this simulation is relevant to the shallow sea environment with the rough seabed and sea surface, which cause multiple reflections of signals before it reaches to destination. Overall, the LAD estimator seems to be preferable in such situations, which do not need any extra parameters to be tuned to achieve good results, but it is more computationally expensive than the other robust estimators.

## 9. Summary and Discussion

In general, location estimation based on range/range-difference measurements is a non-convex problem. In the literature, several computationally expensive approaches are proposed to estimate the accurate location. Instead, we chose two computationally less expensive estimators, namely, the CF and the GN methods, and suggested guidelines to deploy anchors in certain geometrical configurations in the ROI, which guarantees that the two methods can estimate the correct location. We compared the performance of the two methods and suggested when to use one over the other.

Varying sound speed and multipath propagation are important phenomena in underwater acoustic channels that adversely affect the range/range-difference measurements used for localization. In Section 4, we discussed in detail the multipath propagation using both the statistical and deterministic models. Based on the certain common observations found by several researchers in their experiments and the BELLHOP ray tracing simulations, we found that multipath propagation can be fairly modeled by the modified UWB S-V model after certain adaptions. According to this model, an acoustic signal arrives as multiple *eigenpaths*, which contains multiple *eigenrays*. The time arrivals of the *eigenpaths* and the *eigenrays* follow two Poisson distributions with different arrival rates. Based on these observations and other plausible assumptions in Section 5, we approximated the errors in ToA measurements by i.i.d. Exponential distributions, and thus the errors in TDoA measurements by i.i.d. Laplacian distributions, which has possibilities of large values.

As large errors (outliers) in TDoA measurements are possible, a basic multilateration approach (e.g., CF or GN method) is not suitable for robust location estimates. Thus, in Section 6, we proposed three robust location estimators: LAD, LMedS, and RANSAC. The LAD estimator is the direct consequence of the Laplacian distribution of the range-difference measurement errors, whereas we chose the LMedS and RANSAC for their high *breakdown point* and low computational cost. We appropriately used the CF and GN methods all the robust estimators to have low computational expenses. In Section 8, we performed several Monte-Carlo simulations to evaluate and compare the performance of the three robust estimators in term of two metrics: *bias* and *variance*. In simulations, first, we considered the well-studied Gaussian noise in the range-difference measurements, and then we considered the Laplacian noise suggested by the modified UWB S-V model. From the simulation results, we have the following observations and conclusions.

In the case of low-level noise in the inlier range-difference measurements and sufficient separation from the outlier measurements, the RANSAC estimator performed slightly better than the LMedS estimator and significantly better than the LAD estimator. Such situations are relevant to moderate to deepsea environments. On the other hand in presence of high-level noise in the inlier range-difference measurements and not well distinguishable from outlier measurements, the LAD estimator performed slightly better than the other two robust estimators. Such situations are relevant to the shallow sea environments with the rough seabed and wavy sea surface.

From the above general observations, we can draw the following conclusions. For moderately deep to highly deep sea environments, the RANSAC estimator can be a good choice when we have extra information to choose suitable values for the two parameters: τ and *K*. In the worst case, we can choose *K* assuming that there are 50% outliers. The LMedS estimator is the next good choice because it performed only slightly inferior to the RANSAC but at almost half the computational expenses of the RANSAC estimator for the same number of iterations and it needs only one parameter τ to be tuned. When there is not sufficient information to decide a good value of τ, we can drop the final refinement step from the LMedS estimator, and it can still provide a robust location estimate, which can be sufficiently accurate in many practical applications. For the shallow sea environment with the rough seabed and wavy sea surface, the LAD estimator is the preferable choice, which does not need any extra parameter to be tuned to achieve a robust estimate, but it is more computationally expensive than the other two robust estimators.

To perform robust localization of UWASNs, numerous anchors are required to be deployed in the ROI. However, deploying multiple numbers of anchors in the marine environment is not a practically viable option for many practical reasons. Thus, in Section 7, we proposed silent localization using single or multiple surface mobile anchors. We presented three practical schemes with different navigation strategies for the mobile anchor(s) to suit the scale of the UWASNs and the resource available at hand. Combining a mobile anchor scheme with a robust estimator, we proposed a complete package of an efficient, robust, and practically usable localization scheme for low-cost low-power UWASNs.

## 10. Conclusions

In this paper, we proposed robust silent localization schemes for large-scale UWASNs, which is based on a fairly suitable statistical model of the possible errors in the range-difference measurements due to varying sound speed and multipath propagation. To avoid deployment of multiple static anchors in a marine environment, which is practically not suitable in many cases, we proposed three silent localization schemes using a single or two mobile anchors and presented in detail navigation strategies and beacon signaling schedule. Depending upon the noise condition of the acoustic channel, the required accuracy in the estimated locations, and computational resources present in the sensor nodes, we provided guidelines for choosing a robust location estimator and a mobile anchor scheme among the three presented in this paper. In this work, we assumed that the locations provided by a GPS on the mobile anchor are sufficiently accurate. However, this may not be always the case as GPS has a typical accuracy ranging between 10 cm to 150 cm. We leave to our further research on improving the location accuracy due to this. Currently, we are working toward setting up a small-scale UWASN in the North Sea using in-house-developed low-cost and low-power sensor nodes, and we plan to evaluate the performance of the proposed robust localization schemes in the real sea environment.

## Figures and Tables

**Figure 1 sensors-21-00727-f001:**
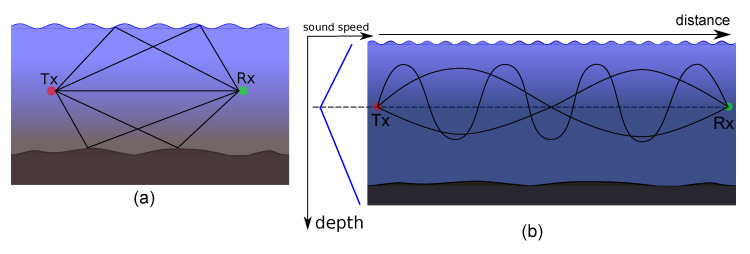
Multipath propagation of acoustic signal: (**a**) in shallow sea for short distance (**b**) in deep sea for long distance. The solid lines between Transmitter (Tx) and Receiver (Rx) are some of the possible rays between them. In shallow sea the sound speed does not change significantly while in deep sea sound speed varies significantly with depth and typically follow Munk profile as in left panel of subfigure (**b**).

**Figure 2 sensors-21-00727-f002:**
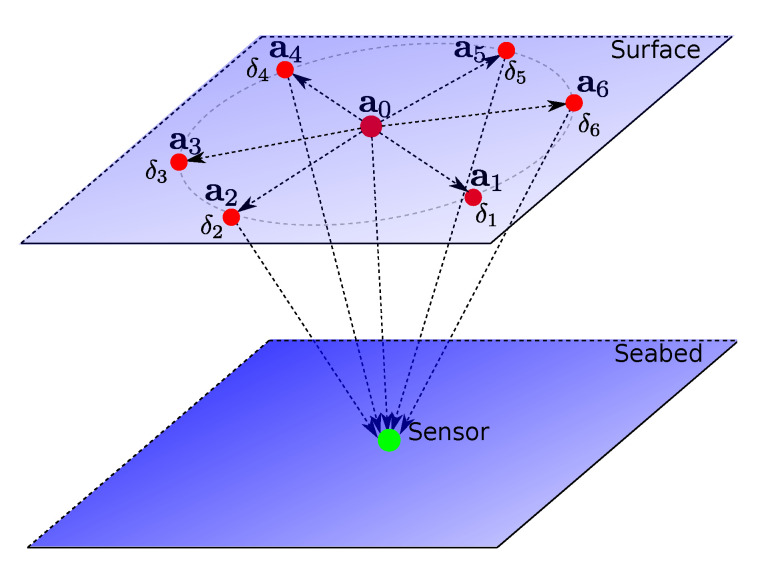
An illustration of the Underwater Positioning Scheme (UPS). Multiple anchors lie on the sea surface and a sensor at seabed. UPS process starts with a broadcast of a beacon signal from the lead anchor with location $a_{0}$, which is then followed by broadcasts from the assistant anchors with locations $a_{i},\forall i=1,\cdots ,6$. A beacon signal contains the tuple $\left(a_{i},\delta_{i}\right)$, where $\delta_{i}$ is the delay between the instant when an assistant anchor receives the beacon signal from the lead anchor and the instance when it broadcasts its own signal. The sensor silently listens to all the signals, records their arrival times and then estimates its location.

**Figure 3 sensors-21-00727-f003:**
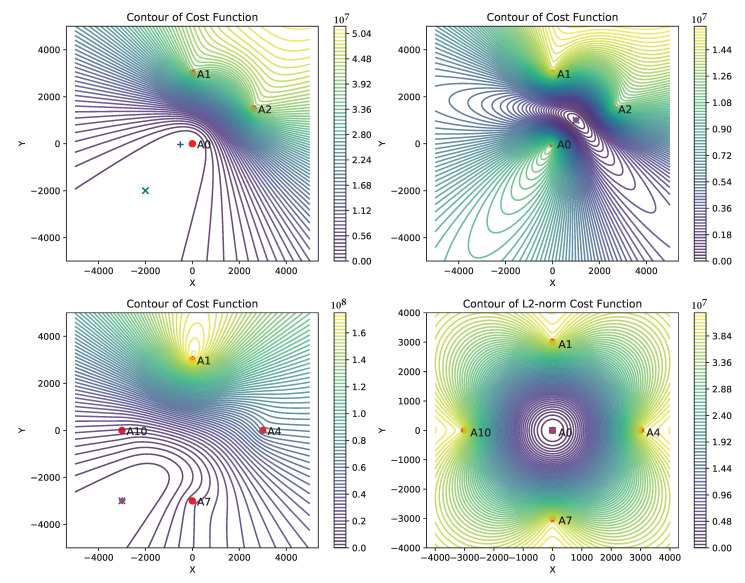
Contour plots of the cost function (Equation 5) for different configuration of the anchors and sensor. X and Y are in meters. The plots compares the configurations that are easier to find the correct solution by an iterative solver. A0 represent the lead anchor and other red discs represent assistant anchors. The marker x represents actual location of the sensor and +, and ∗ represent estimated locations by the GN and CF methods, respectively. In all plots the estimated locations by the GN and CF methods are close to the true locations except in the top-left plot where the solution does not exits for CF method.

**Figure 4 sensors-21-00727-f004:**
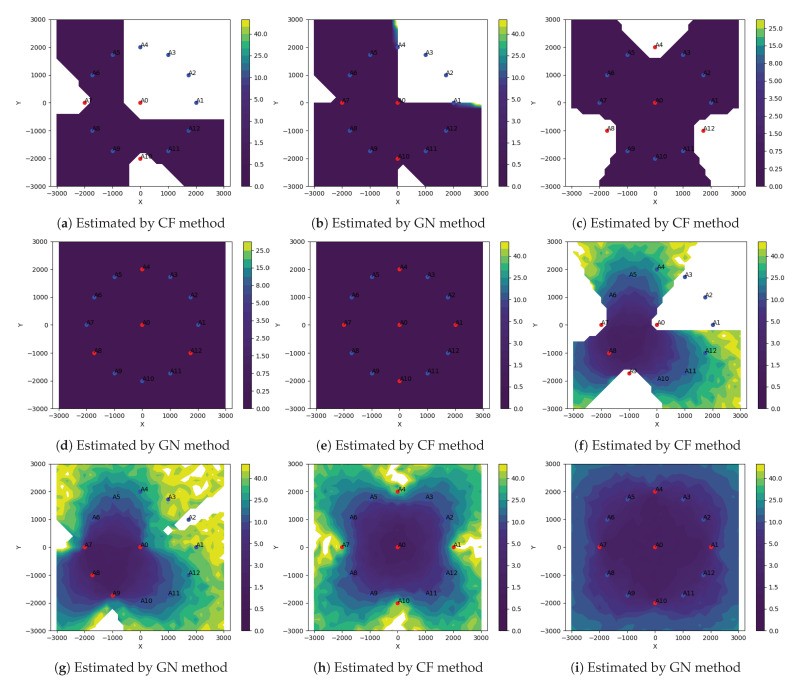
Comparison between the CF and the GN methods based on uniquely localizable areas. The plots show maps of *mean-L2-norm-error* calculated over 100 trials for each sensor located at 100 m depth on a 2D rectangular grid of size 31 × 31 over a region of 6000 × 6000 m^2^. The discs (blue and red) represent the anchors placed symmetrically on a circle with radius = 2000 m and center at origin. The red discs are the anchors which take part in TDoA measurements and the name with lowest index represents the lead anchor. Subfigures (**a**–**e**) show the localizable area when no noise in *t_i_* measurements, whereas subfigures (**f**–**i**) are due to i.i.d. Gaussian noise with *σ* = 1.0 ms in *t_i_* The dark purple are regions where sensor are localized uniquely and white regions are where either the *mean-L2-norm-errors* are very large or no solution is found. Note that in all the simulations the sound speed is *v* = 1530 m/s. The plot clearly shows that GN method localizes larger area with lower errors than the CF method.

**Figure 5 sensors-21-00727-f005:**
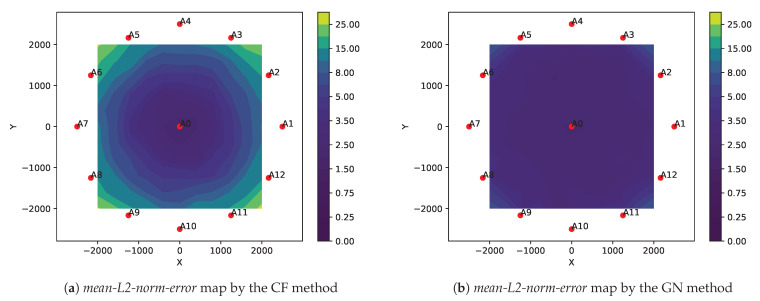
Comparison between the CF and GN methods based on estimation errors when all the sensors are surrounded by anchors. In this simulation, sensors are placed at 100 m depth on a rectangular grid of size 11 × 11 over region of 4000 × 4000 m^2^. The assistant anchors are deployed on the surface symmetrically on a circle with radius of 2000 m and the lead anchor at origin. Plots (**a**,**b**) show the *mean-L2-norm-errors* map calculated over 100 trials when there are i.i.d. Gaussian noise with *σ* = 1.0 ms in *t_i_* measurements. The plots clearly show that the GN method produces significantly lower estimation errors over larger area than the CF method.

**Figure 6 sensors-21-00727-f006:**
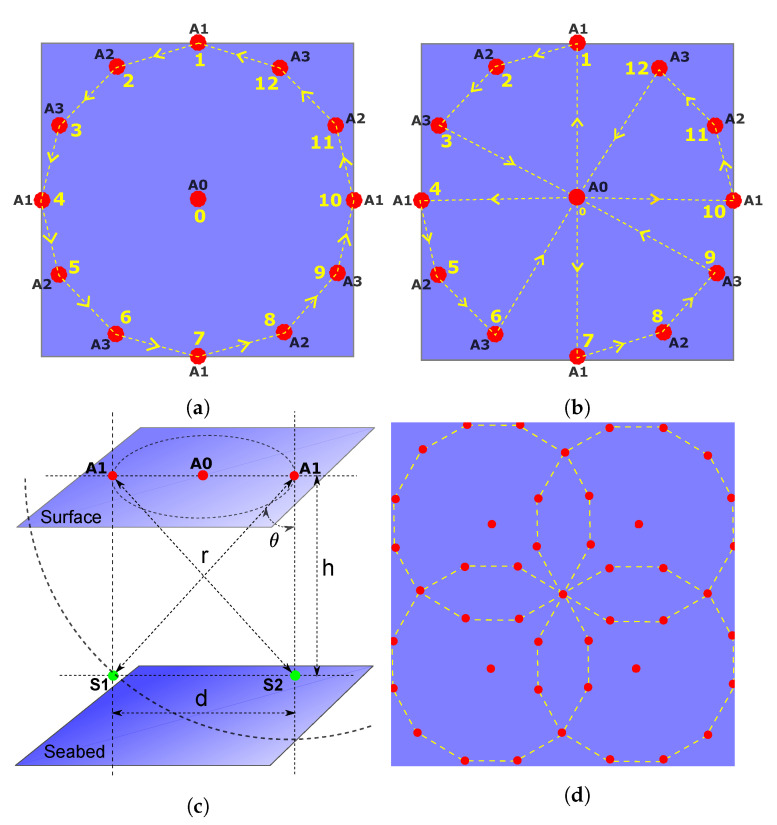
Geometric configurations of a stationary and/or a mobile anchor for silent localization: (**a**) a stationary gateway at the center acts as a lead anchor (**A0**) while a mobile anchor moves on a circular path to different positions marked as **1**, ⋯, **12** acting as assistant anchors (**A1**–**A3**), and repeating the pattern. (**b**) A single mobile anchor acting both as the lead (**A0**) and the assistant anchors (**A1**–**A3**) at a time while moving in the path shown by dashed line with arrows. (**c**) A 3D geometric of a network where anchors (A0,A1) are on sea surface and the sensors (**S1**,**S2**) are lying at depth *h* below the surface. (**d**) A large region-of-interest (ROI) is divided into overlapping cells with positions of the lead and assistant anchors.

**Figure 7 sensors-21-00727-f007:**
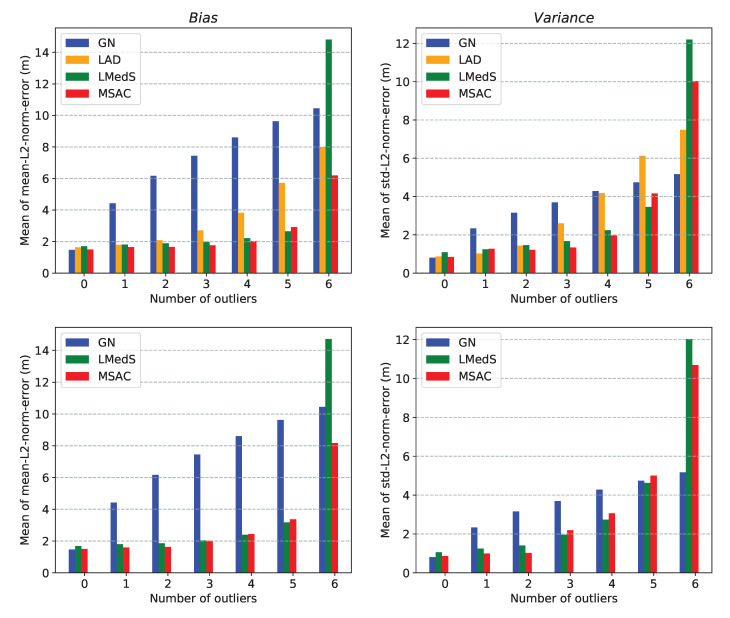
Simulation results for zero-mean i.i.d. Gaussian noise in the ToA measurements. First column show *bias* and second column show the *variance*. First row shows the result for σ=1.0 ms and T=120, and second row for σ=1.0 ms and T=35. Third row shows the result for σ=2.0 ms. The outliers are chosen from an i.i.d. Uniform distribution in the interval [−30,−10]⋃[10,30] ms.

**Figure 8 sensors-21-00727-f008:**
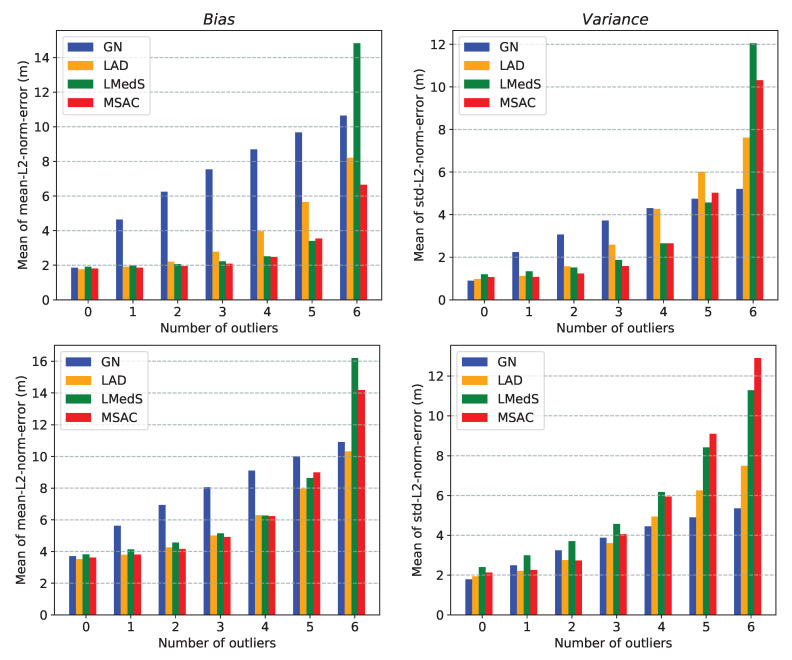
Simulation results for i.i.d. Exponential noise in the ToA measurements. First column shows *bias*, and second column shows *variance*. The first and second row shows the results when the inlier ToA measurements have Exponential noise with scales σ=1.0 ms and σ=2.0 ms, respectively. The outliers are created by adding random values from an i.i.d. Uniform distribution in the interval [−30,−10]⋃[10,30] ms to some of the ToA measurements.

**Figure 9 sensors-21-00727-f009:**
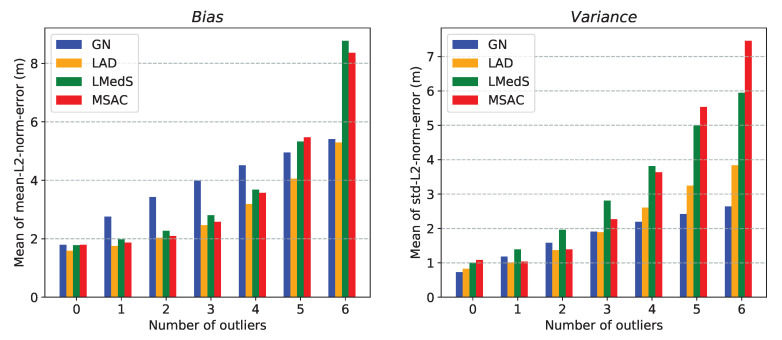
Simulation results for i.i.d. Exponential noise in the ToA measurements with distance-dependent scale. The first column shows the *bias*, and the second column shows *variance*. The noise in ToA measurements are from an i.i.d. Exponential distribution with scale σ=d/3000 ms, where *d* is the actual distance (m) between a transmitter and a receiver. The outliers are created by adding random values from an i.i.d. Uniform distribution in the interval [−15,−5]⋃[5,15] ms to some of the ToA measurements.

**Table 1 sensors-21-00727-t001:** Comparison between the CF and the GN method with varying noise level and number of anchors. Table (**a**) shows the effect of increasing levels of i.i.d. Gaussian noise in $t_{i}$ measurements on the *bias* and the *variance*. In this case, 12 assistant anchors are take part in Time-Difference-of-Arrival (TDoA) measurements. Table (**b**) shows the effect of increasing number of anchors involved in localization on the two metrics. The assistant anchors are deployed symmetrically on a circle around the lead anchor at the origin and the noise in $t_{i}$ measurements are i.i.d. Gaussian with σ=2.0 ms. The geometric configuration of the anchors and the sensors are shown in Figure 5.

(a)					
	**STD (in ms**)				
Errors in $t_{i}$	0.001		0.002		0.003
Methods	*bias* (in m)				
GN	1.6612		3.3226		4.9842
CF	4.4564		8.9129		13.3698
	*variance* (in m)				
GN	0.9548		1.9098		2.8651
CF	3.0070		6.0148		9.0237
(**b**)					
	**Numbers of Anchors**				
	4	7	10	13	16
Methods	*bias* (in m)				
GN	5.8692	4.1281	3.6393	3.3226	3.2494
CF	14.3327	11.1886	9.78817	8.9129	8.4743
	*variance* (in m)				
GN	3.3469	2.2586	2.0275	1.9098	1.9071
CF	9.5370	7.5355	6.5978	6.0148	5.8165

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
