# Peer review of "Robust Silent Localization of Underwater Acoustic Sensor Network Using Mobile Anchor(s)"

_sensors, 2021, doi:10.3390/s21030727_

Round 1
Reviewer 1 Report
The authors proposed a Robust Silent Localization of Underwater Acoustic Sensor Network Using Mobile Anchor. The proposed three robust estimators are evaluated to show the effectiveness of the proposed design. However, the work needs improvement to incorporate the following concerns and comments.
1. The authors proposed three different configurations of multiple mobile anchors and analyzed how they work, but the author did not analyze the performance of the three schemes. As one of the main contributions of this paper, such a result is not convincing.
2. When presenting the "Underwater Acoustic Channel Modelling", The author compares different documents and believe that the modified ultra-wideband Saleh-Valenzuela (UWB S-V) model is a fairly suitable for characterizing acoustic channel fading, but I can’t see the authors’ contribution in this part.
3. Multipath effect is an important problem to be solved in this article. When modified UWB S-V model is applied to underwater acoustic channels. How does it mitigate the effects of multipath? And this part lacks comparison and simulation results.
4. The propagation delay of sound in the water is large, so the mobility of the node has always been a great interference to the underwater acoustic localization. Will the mobile anchor model in this paper affect the positioning accuracy?
5. “The nodes only listen” means that the target node has been waiting for the signal, which will produce greater energy consumption. How to solve this problem?
The simulation results section is one of the most important sections of the paper and should be significantly improved according to the comments above. Many details and discussions are currently missing making this section quite poor. Although the objectives of this work look clear, the overall contribution seems weak.
Reviewer 2 Report
I really like the topic and ideas in the paper. The main issue I am facing is the write-up and presentation, which I find rather confusing (I think I lost track of what's going on somewhere on page 9). It seems that the paper tries to make several contributions and thus becomes very scattered and looses focus. Moreover, it is unclear what contributions have been made in particular, and what concepts have been used rather than invented. This is mainly due to the fact that there is not one concise related work section, but related work is sprinkled over the paper and modifications and additions by the authors are introduced on-the-fly. Hence, I found it difficult to read and appreciate the paper.
As an example, the explanation of GN and CF are rather fuzzy and the figures do not help to track down what the advantages and disadvantages are. I see no pattern in the comparison. Thus, I can't really draw the general insight/meaning from Fig. 3, because the located sensor is jumping around from subfigure to subfigure and anchor choices appear random. What is the authors' idea behind the shown setup? Figure 4 is way to small and the arrangement appears rather random. Please increase figure (and font size in particular) and rearrange. I also wonder why the short math part for CF and GN have been moved to the appendix.
In Sect. 4, it is a bit unclear to me why the authors look into this issue. Is this because it is required to analyze localization performance in more detail? I am somewhat missing the link (or central theme). The entire section feels a bit odd to.
Only after 25 (of 28 content) pages, a concise evaluation is presented. I would have liked to see a deeper and extended study of parameters and influencing factors. And it probably would make sense to split the paper somewhere or at least write it in a more structured way.
minor comments:
- l 178: scaler -> scalar?
- l 179: are by -> by
- page 6 second paragraph, sentence starting with "The Gaussian distribution is not" is doubled
- l 216: a unknown -> an unknown
- the markers x,+, are hard to identify in Fig. 3 (I cannot even spot in the top left one). What are the units of x, y, and z (contour lines)?
- the different y-scales of the evaluation pots make a comparison difficult
Round 2
Reviewer 1 Report
The response of the authors sounds good.
Author Response
We are highly thankful for your insightful suggestions/comments that helped to improve the quality of the paper.
We have carefully gone through the draft to check the English language and spell-check.
Reviewer 2 Report
The authors have improved the paper considerably. There remain some minor questions and aspects that could be improved, mainly the evaluation is still rather short and not as strong as it could be. However, I support acceptance of the paper.
Minor suggestions:
- Use some y-axis scaling in Figs. 7 - 9 to improve comparability of results
- Spell-check and check grammar
Author Response
We are highly thankful for your insightful suggestions/comments that helped to improve the quality of the paper. Below are our responses to your comments.
Regarding the evaluation of the performance of the mobile anchor schemes, we believe that it can be better evaluated only by conducting real sea trials, which we plan to do in the near future.
Regarding the y-axis scaling in Figs. 7-9, we tried to bring the plots in the same scale, however, doing so makes the minor differences among the bars not visible. Thus, we preferred to keep them unchanged.
We have carefully gone through the draft to check the English language and spell-check.